# The Generalization Spectrum: A Chromatographic Approach to Evaluating Learning Algorithms

**Jinghan Zhang** [1 2 †]  **Zerui Cheng** [3 *]  **Shiqi Chen** [4]  **Ge Zhang** [1]  **Wenhao Huang** [1]  **Jiashuo Liu** [1]  **Junxian He** [2]
**Tianle Cai** [1 †]

## Abstract

Traditional evaluations measure a learning algorithm's final performance on an i.i.d. test set, reducing learning to a single aggregate score. This approach obscures a fundamental question: to what extent does learning from a specific example generalize to others? Such per-sample generalization, akin to learning by analogy in human cognition, captures how far the knowledge extracted from one example can transfer, yet remains invisible to standard benchmarks. We introduce the Generalization Spectrum, an evaluation framework designed to expose this hidden dimension. For each training example, we construct a controlled suite of test variants arranged by increasing transfer distance, from exact recall to implementation transfer across languages, context transfer under complete narrative re-framing, category-matched in-domain problems, and an unpaired baseline. By tracking performance across these distances, we reveal not just whether an algorithm learns, but how far that learning extends. We instantiate this framework on competitive programming, using a selection-and-synthesis pipeline seeded with recent problems to mitigate contamination. We first compare **three canonical learning paradigms** under matched memorization. RL converts memorization into near-transfer more efficiently than SFT-family baselines, while ICL exhibits strong but correspondence-dependent transfer. We then use the Spectrum to diagnose **within-family variants**. The resulting profiles show that local gains need not expand the generalization radius: abstractions and hints mainly lift local

transfer, RFT preserves a stronger far-transfer tail than reference SFT, and self-distillation or hint-assisted RL can reduce far transfer even when local transfer or optimization improves.

## 1. Introduction

Post-training methods—prompting, supervised fine-tuning, and reinforcement learning—can achieve comparable aggregate scores on standard benchmarks, yet produce qualitatively different behaviors when probed more carefully (Kirk et al., 2024; Chu et al., 2025). A model that "passes" a benchmark may be memorizing training instances, transferring under minor perturbations, or acquiring abstractions that survive substantial distributional shifts—but aggregate metrics conflate these distinct capabilities. Recent work has begun to compare paradigms on in- vs. out-of-distribution splits (Wang et al., 2024), but such binary comparisons still obscure a key question: how far does learning generalize from each training example?

We introduce the Generalization Spectrum, a framework that evaluates generalization as a function of transfer distance rather than a binary in/out split. The core idea is to construct paired test variants derived from the same seed instance, ordered by increasing distance from the original. Evaluating a model across these levels yields a Generalization Profile—a curve that reveals distance-dependent decay patterns invisible to any single score (Kaplan et al., 2020). We instantiate five levels: exact recall of training instances (D0), implementation transfer across programming languages (D1) (Cassano et al., 2023), context transfer under complete narrative reframing with preserved mathematical structure (D2), category-matched problems sharing algorithmic tags (D3), and an unpaired in-domain baseline (D4). This design enables fine-grained diagnosis: two methods with identical aggregate performance may exhibit strikingly different profiles—one decaying sharply after D1, another maintaining transfer through D2.

A central challenge in comparing learning paradigms is that stronger training tends to increase both memorization and transfer simultaneously (Power et al., 2022; Nakkiran et al., 2020). If method A outperforms method B on trans-

*Work done while at Seed, ByteDance. †Corresponding authors. [1]ByteDance Seed [2]Hong Kong University of Science and Technology [3]Princeton University [4]University of Oxford. Correspondence to: Jinghan Zhang <zhangjinghan.23@bytedance.com>, Tianle Cai <caitianle@bytedance.com>.

*Proceedings of the $43^{rd}$ International Conference on Machine Learning*, Seoul, South Korea. PMLR 306, 2026. Copyright 2026 by the author(s).

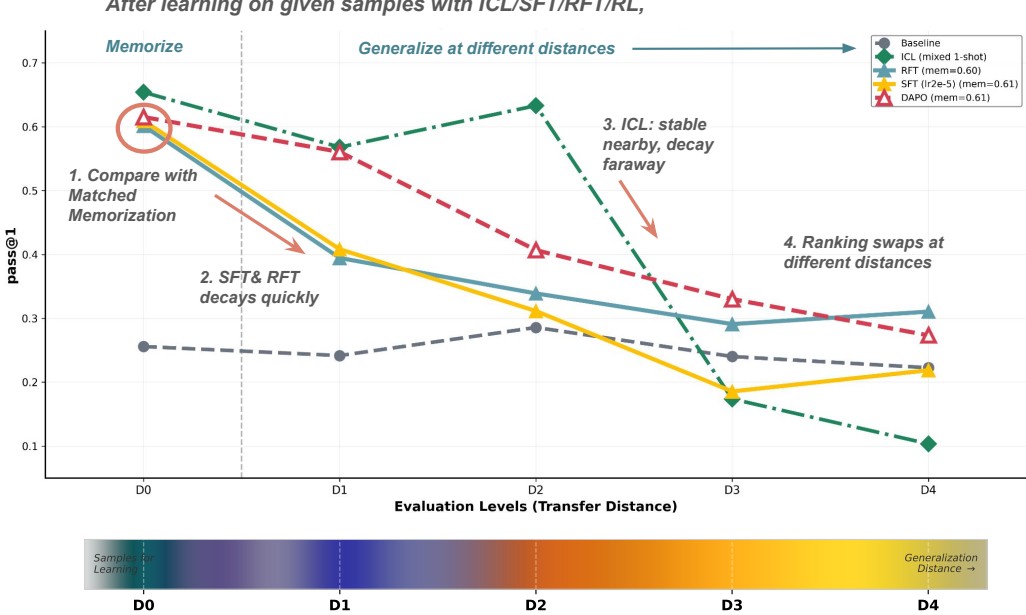

*Figure 1.* Generalization Spectrum profiles for SFT, GRPO, and ICL. At matched memorization, GRPO yields stronger near-transfer than SFT, especially on implementation and context transfer. Paired ICL remains strong through the instance-paired levels ($D_0$–$D_2$) but drops at category-matched and unpaired levels ($D_3$–$D_4$), exposing a correspondence bottleneck rather than uniformly stable transfer.

fer tasks but also shows higher training-set performance, is the transfer gain genuine or merely a side effect of better memorization? To isolate transfer efficiency—the ability to convert memorization into generalization—we propose matched-memorization comparison: selecting checkpoints where different methods achieve comparable D0 (exact recall) performance, then comparing their behavior at D1 and beyond. This controlled setup enables principled investigation along two practical axes: how to learn (the choice of learning algorithm) (Mosbach et al., 2023) and what to learn (the content used for adaptation) (Zhou et al., 2023).

We instantiate the Generalization Spectrum on competitive programming (Li et al., 2022; Hendrycks et al., 2021; Jain et al., 2024), a domain particularly suited for this investigation. First, correctness is unambiguous: solutions either pass all test cases or fail, eliminating evaluation subjectivity (Chen et al., 2021). Second, the domain supports controlled paired variants: a single algorithmic problem can be expressed in different languages (D1) (Cassano et al., 2023; Peng et al., 2024), reframed with entirely different narratives while preserving mathematical structure (D2), or matched to problems requiring similar algorithmic techniques (D3). Third, competitive programming platforms continuously release new problems, providing fresh evaluation material as models evolve (Jain et al., 2024). We construct a benchmark of 256 evaluation instances (64 seed problems × 4 spectrum levels), with D2 variants generated through a multi-stage pipeline involving cross-model generation and verification

(details in Appendix C).

Using matched-memorization comparison, we first compare three canonical paradigms—ICL (Brown et al., 2020; Dong et al., 2024), SFT, and RL (Ouyang et al., 2022; Shao et al., 2024)—across the Generalization Spectrum. The comparison reveals distinct decay profiles under matched seed recall: reference-solution SFT concentrates its gains near implementation transfer and largely exhausts them after narrative recontextualization, GRPO attenuates with distance but extends its gains through the far tail of the Spectrum, and paired ICL remains strong across instance-paired levels before dropping once direct demonstration-target correspondence is removed. More strikingly, D2 exposes divergent transfer mechanisms: gradient-based imitation shows sharp degradation under narrative recontextualization, while ICL maintains strong transfer—suggesting these paradigms exploit fundamentally different generalization pathways (Mosbach et al., 2023). We then use the Spectrum to diagnose within-family variants—ICL content variants (abstraction, hints), SFT target-source variants including RFT (Yuan et al., 2023), self-taught distillation (SDFT (Shenfeld et al., 2026; Zhao et al., 2026; Hübotter et al., 2026)), and a coding-adapted hint-assisted GRPO variant inspired by sparse-reward guidance methods (Li et al., 2026; Liu et al., 2025b; Zhang et al., 2026; 2025; Wang et al., 2026; Huang et al., 2025a; Kang et al., 2025)—asking whether they expand the generalization radius or merely shift local height, speed, or near-far concentration.

*Table 1.* Generalization Spectrum. Columns mark which attributes are intentionally shared with the training seed. Narrative excludes the formal I/O contract; Exec. Spec. abbreviates executable specification and includes the I/O contract, solution logic, and tests. Moving downward removes one intended shared attribute at a time.

| Level | Name | Format | Narrative | Exec. Spec. | Family |
|-------|------|--------|-----------|-------------|--------|
| D0 | Exact Recall | ✓ | ✓ | ✓ | ✓ |
| D1 | Impl. Transfer | ✗ | ✓ | ✓ | ✓ |
| D2 | Context Transfer | ✗ | ✗ | ✓ | ✓ |
| D3 | Category Matched | ✗ | ✗ | ✗ | ✓ |
| D4 | Unpaired Baseline | ✗ | ✗ | ✗ | ✗ |

In this work, we propose the Generalization Spectrum, a distance-aware evaluation framework with paired transfer levels (D0–D4) and profile-based analysis. We introduce matched-memorization comparison to isolate transfer efficiency across learning paradigms, and construct a competitive-programming benchmark with 64 seeds and 256 evaluation instances. Using this framework, we compare vanilla ICL, SFT, and RL under matched memorization to characterize their generalization profiles, and further diagnose recent within-family variants—ICL content design, SFT target-source/RFT, SDFT, and hint-assisted GRPO. Our findings establish that algorithm choice reshapes the generalization profile, that ICL and gradient-based methods preserve different forms of correspondence, and that within-family variants can lift local transfer, smooth the supervised tail, write ICL-style local gains into parameters, or accelerate optimization while compressing far-transfer behavior.

## 2. The Generalization Spectrum Framework

Traditional evaluation reduces a learning algorithm's performance to a single aggregate score, obscuring a key question: *how far does learning generalize?* We introduce the **Generalization Spectrum**, an evaluation framework that exposes this hidden dimension. The core idea is to construct paired test variants arranged by increasing transfer distance—from exact recall through surface perturbation to category-matched and domain-level problems—thereby characterizing the *generalization radius* of learning rather than merely final performance. Figure 1 illustrates the key insight: plotting pass rate as a function of transfer distance yields a **Generalization Profile** that reveals how different learning algorithms (ICL (Brown et al., 2020), SFT (Ouyang et al., 2022), RFT (Yuan et al., 2023), RL (Schulman et al., 2017; Shao et al., 2024; Yu et al., 2025)) exhibit distinct decay patterns.

### 2.1. Spectrum Levels

We define transfer distance by which pieces of information remain shared between a seed instance and its evaluation variant. We consider four attributes: (i) implementation format, the surface realization of the solution; (ii) problem narrative, the natural-language context of the task excluding the formal I/O contract; (iii) executable specification, including the reference algorithm, I/O contract, and test suite; and (iv) problem family, the broader algorithmic category.

Table 1 gives a nested spectrum over these attributes. Moving from D0 to D4 removes one additional shared attribute: D0 measures exact recall, D1 changes only the implementation format, D2 additionally changes the narrative while preserving the executable task, D3 keeps only the problem family, and D4 removes intentional seed-specific pairing. We use D0–D4 as the transfer-distance axis throughout the paper.

### 2.2. Benchmark Instantiation and Construction

We instantiate the abstract spectrum in competitive programming (Hendrycks et al., 2021; Li et al., 2022) because the domain offers (i) unambiguous correctness via execution against test cases, (ii) a rich space for controlled variation around a single algorithmic idea, and (iii) a continuously growing problem pool that guards against data contamination (see Appendix C.1 for further discussion).

To operationalize the spectrum in this domain, we map the abstract attributes in Table 1 to concrete competitive-programming artifacts. A seed problem is represented by a problem narrative, a formal I/O contract, reference tests, a reference solution, and algorithmic tags. The implementation format corresponds to the requested programming language and surface code form; the narrative is the natural-language task context; the executable specification consists of the I/O contract, reference algorithm, and test suite; and the family is the algorithmic tag or category.

Given this decomposition, we instantiate the five spectrum levels for each seed problem as follows:

- **D0: Exact recall.** The original problem paired with its Python reference solution.

- **D1: Implementation transfer.** The same problem, I/O contract, tests, and solution logic, but evaluated in C++.

- **D2: Context transfer.** A newly narrated problem that preserves the same executable specification and solution logic.

- **D3: Category-matched transfer.** A different coding problem selected to share the seed's algorithmic family.

- **D4: Unpaired baseline.** A recent in-domain problem sampled without enforcing any tag or structural relationship to the seed.

Following the construction pipeline detailed in Appendix C, we build the **Generalization Spectrum Benchmark** around

64 seeds, yielding 256 transfer instances across D1–D4 while retaining the original seeds as D0 exact-recall references. Worked examples are provided in Appendix D. The framework is model- and source-agnostic, allowing practitioners to instantiate new benchmarks as models evolve.

**Quantitative validation.** The spectrum ordering reflects progressive removal of shared structure rather than an *a priori* assumption about task difficulty. The final two columns of Table 2 provide independent validation against D0: statement-level cosine similarity (Reimers & Gurevych, 2019) decreases monotonically from D1 to D4, and algorithmic tag overlap drops from 100% (D1–D3) to 17.2% at D4 (incidental). The sharp similarity drop from D1 to D2 ($1.000 \rightarrow 0.513$) supports that our recontextualization pipeline produces substantially different problem descriptions rather than paraphrases. D3 and D4 exhibit comparable text similarity (0.373 vs. 0.346) because both use unrelated statements; their distinction is purely structural, residing in shared algorithmic family.

### 2.3. Metrics

Our base metric is **pass@1** ($r_i$) (Chen et al., 2021): the pass rate at each distance level $i$. Let $M$ denote the base model and $M_S$ the model after learning on seed set $S$ (via ICL, SFT, RFT, or RL). Building on this, we define:

- **Gain:** $\text{Gain}(i) = r_i(M_S) - r_i(M)$, the improvement over the base model at spectrum level $D_i$. $\text{Gain}(0)$ measures memorization (Arpit et al., 2017); $\text{Gain}(i)$ for $i \geq 1$ measures transfer to variants at distance $i$.

- **Normalized Gain:** $\text{Gain}_n(i) = \text{Gain}(i)/(1 - r_i(M))$, which accounts for remaining headroom at each distance.

- **Area Under Spectrum (AUS):** Aggregate score computed as $\frac{1}{4} \sum_{i=1}^{4} \text{Gain}(i)$, the average absolute pass@1 gain across $D_1$–$D_4$, enabling cross-method comparison.

- **Normalized Near-Far Gap (N-F$_n$):** N-F$_n$ = $\frac{\text{Gain}_n(1) + \text{Gain}_n(2)}{2} - \frac{\text{Gain}_n(3) + \text{Gain}_n(4)}{2}$, the gap between normalized near-transfer gains ($D_1$–$D_2$) and normalized far-transfer gains ($D_3$–$D_4$). Larger values indicate that transfer gains concentrate near the seed rather than extending to farther variants; when $\text{Gain}_n(i)$ is reported as a percentage, N-F$_n$ is reported in percentage points (Barnett & Ceci, 2002).

## 3. Experimental Setup

We evaluate three canonical learning paradigms, in-context learning (ICL) (Brown et al., 2020; Dong et al., 2024), supervised fine-tuning (SFT), and reinforcement learning (RL), on the Generalization Spectrum using Qwen3-4B-Thinking. For each paradigm, adaptation starts from the same 64 seed

problems and performance is measured across $D_0$–$D_4$ with pass@1 (Chen et al., 2021). Our evaluation follows a four-stage pipeline.

**Stage 1: Apply learning paradigms.** We instantiate each paradigm on the training set. ICL conditions each evaluation problem on its oracle-paired seed demonstration, with a random-demonstration control. SFT fine-tunes on reference reasoning traces and code (Ouyang et al., 2022). RL uses binary outcome reward and reports GRPO (Shao et al., 2024) as the primary algorithm, with DAPO variants (Yu et al., 2025; Schulman et al., 2017) in Table 12. Full training details are in Appendix E.1.

**Stage 2: Select matched checkpoints during training.** To compare methods fairly, we periodically evaluate gradient-based checkpoints on $D_0$ (exact recall of the seed problems) and compare methods only at matched $D_0$ pass@1 levels. The key is to match methods on $D_0$ before interpreting transfer on $D_1$–$D_4$. Final-checkpoint or compute-matched comparisons would otherwise conflate *how much* a method memorizes with *how far* that memorization transfers. ICL has no training trajectory, so it is included at the memorization level induced by its demonstration strategy.

**Stage 3: Evaluate selected checkpoints across the spectrum.** Each selected checkpoint, or ICL-conditioned model, is evaluated across all five levels of the Generalization Spectrum, producing $64 \times 5 = 320$ instances, including 256 transfer instances at $D_1$–$D_4$. We report pass@1 averaged over 8 independent runs using the same decoding and sandbox settings for all methods; prompts and implementation details are in Appendix E.2.

**Stage 4: Compute transfer profiles.** Using the base model as baseline, we compute $\text{Gain}(i)$, normalized gains $\text{Gain}_n(i)$, aggregate scores (AUS), and the normalized Near-Far Gap (N-F$_n$; §2.3). These metrics summarize each method's behavior across $D_0$–$D_4$ as a complete generalization profile.

By fixing the seed set, matching trainable methods on $D_0$ before evaluating $D_1$–$D_4$, and placing ICL at its demonstration-induced $D_0$ level, the pipeline isolates each paradigm's *transfer efficiency*: how effectively a fixed level of seed memorization becomes generalization at increasing transfer distances.

## 4. Comparing Learning Paradigms: How Algorithm Transfer across the Spectrum?

The Generalization Spectrum measures how far learning extends, but this distance depends critically on how learning

*Table 2.* Dataset statistics and transfer-distance validation for each level of the Generalization Spectrum. D0 and D1 share the same problem set but evaluate solutions in different programming languages. D4 covers 23 of the 27 seed categories as it is sampled without category-matching constraints.

| Level | Dataset statistics | | | | | Distance to D0 | |
|-------|-------------------|---|---|---|---|----------------|---|
|       | #Problems | Avg. Rating | Rating Range | Avg. Length | Categories | Cosine Sim. | Tag Overlap |
| D0 | 64 | 1706 | 900–2600 | 487 | 27 | – | – |
| D1 | 64 | 1706 | 900–2600 | 487 | 27 | $1.000 \pm .000$ | 100% |
| D2 | 64 | 1736 | 1236–2450 | 416 | 27 | $0.513 \pm .086$ | 100% |
| D3 | 64 | 1905 | 900–2900 | 448 | 27 | $0.373 \pm .095$ | 100% |
| D4 | 64 | 2076 | 1100–3200 | 514 | 23 | $0.346 \pm .096$ | 17.2% |

happens. Prior comparisons suggest that paradigm choice can change generalization even when data or adaptation budgets are controlled (Mosbach et al., 2023; Chu et al., 2025). Given identical training content (the same 64 seed problems), do ICL, SFT, and RL produce the same generalization profile? Or does algorithm choice reshape the spectrum itself? We compare these paradigms under matched-memorization conditions: checkpoints are selected where $D_0$ performance is equivalent, isolating each algorithm's transfer efficiency, its ability to convert memorization into generalization at increasing distances.

### 4.1. Main Results

Table 3 presents performance across the Generalization Spectrum at matched memorization levels. A key pattern validates the spectrum design: RL gains generally attenuate as transfer distance increases, yet remain positive across all levels, while SFT drops sharply at far distances. At matched $D_0 \approx 0.6$, RL improves over SFT on near transfer ($\text{Gain}_n(D_1)$ of 39.0% vs. 21.9%; $\text{Gain}_n(D_2)$ of 13.6% vs. 3.0%) and preserves positive far-transfer gains ($\text{Gain}_n(D_3)$/$\text{Gain}_n(D_4)$ of 12.3%/16.8% vs. $-7.1\%$/$-0.5\%$ for SFT). Thus, outcome-based training not only strengthens near transfer but also avoids the far-distance collapse observed under reference-solution SFT.

### 4.2. Which Paradigm Best Converts Memorization into Transfer?

**ICL: strong but fragile context-based transfer.** Paired ICL shows impressive gains across $D_0$–$D_3$ when the oracle retriever provides the matched seed example. However, random ICL performs *worse* than the base model, highlighting the brittleness of context-based adaptation: its effectiveness hinges entirely on demonstration selection quality.

**RL vs. supervised imitation.** At matched $D_0 \approx 0.6$, RL outperforms reference SFT at every transfer level: $D_1$ (0.54 vs. 0.41), $D_2$ (0.38 vs. 0.31), $D_3$ (0.33 vs. 0.19), and $D_4$ (0.35 vs. 0.22). The advantage is not confined to implementation or context transfer; it remains visible at far distances

where SFT's normalized gains become negative or near zero. This suggests that outcome-based training learns a transfer profile that attenuates with distance but does not collapse as sharply as supervised imitation.

$D_2$ **reveals divergent transfer mechanisms.** At $D_2$ (Context Transfer), where problems are recontextualized with different narratives, we observe a striking divergence: reference SFT shows a sharp drop ($\text{Gain}_n(2)$ of 3.0% at matched $D_0 \approx 0.6$), while ICL maintains strong transfer ($\text{Gain}_n(2)$ of 48.6%), even exceeding its $D_1$ performance. RL lies between these regimes, retaining a positive $D_2$ gain (13.6%) but still attenuating relative to $D_1$. This asymmetry suggests that context-based learning leverages demonstration-test correspondence most effectively under surface variation, while parameter-based methods differ in how much of the underlying structure they preserve.

### 4.3. Inspecting Failure Modes behind the Spectrum

Aggregate pass rates show where the learning paradigms separate, but they do not reveal which errors drive the separation. We therefore use $D_1$ and $D_2$ as two controlled diagnostic lenses. $D_1$ changes the implementation language while keeping the problem statement fixed, so the model can still identify the seed problem from its remembered surface form. $D_2$ is farther: the narrative is recontextualized while the executable task is preserved, so surface memorization of the original statement no longer identifies the problem. This setting isolates whether training preserves solution-family identification from the underlying structure rather than from the original wording.

On $D_1$, the main gap is an implementation-language failure: among various failure categories, **compile errors surge specifically in SFT, while RL maintains stable compilation rates**. As shown in Figure 2a, SFT maintains an average compile error rate of 17.6%, whereas RL remains stable at 5.5% throughout training. This pattern is diagnostic: SFT learns by imitating *surface patterns* of Python solutions, which corrupt cross-lingual transfer. In contrast, RL, trained via outcome reward, avoids imitating reference-

*Table 3.* Performance across the Generalization Spectrum. Each cell shows pass@1 (bold) with Gain($i$) over the base model as subscript ($\uparrow$ positive, $\downarrow$ negative). $D_0$ measures seed memorization; $D_1$–$D_4$ measure transfer to increasingly distant variants. **Gain$_n$** normalizes each level's gain by remaining headroom. **AUS** averages raw transfer gains across $D_1$–$D_4$, while **N-F$_n$** measures whether normalized gains concentrate on near-transfer ($D_1$–$D_2$) rather than far-transfer ($D_3$–$D_4$), reported on the same percentage-point scale as Gain$_n$. See §2.3 for formal definitions.

| Method | $D_0$ (Exact Recall) Result | Gain$_n$ | $D_1$ (Impl. Transfer) Result | Gain$_n$ | $D_2$ (Context Trans.) Result | Gain$_n$ | $D_3$ (Category Match) Result | Gain$_n$ | $D_4$ (Unpaired) Result | Gain$_n$ | AUS | N-F$_n$ |
|---|---|---|---|---|---|---|---|---|---|---|---|---|
| Base model | $0.26_{+0.00}$ | — | $0.24_{+0.00}$ | — | $0.29_{+0.00}$ | — | $0.24_{+0.00}$ | — | $0.22_{+0.00}$ | — | 0.00 | 0.0% |
| ICL (paired) | $0.65_{\uparrow+.40}$ | +53.5% | $0.57_{\uparrow+.33}$ | +43.0% | $0.63_{\uparrow+.35}$ | +48.6% | $0.17_{\downarrow-.07}$ | -8.7% | $0.10_{\downarrow-.12}$ | -15.4% | +0.10 | +57.9% |
| ICL (random) | $0.21_{\downarrow-.05}$ | -6.6% | $0.17_{\downarrow-.07}$ | -9.5% | $0.21_{\downarrow-.07}$ | -10.0% | $0.06_{\downarrow-.06}$ | -6.3% | $0.10_{\downarrow-.12}$ | -15.4% | -0.06 | +1.1% |
| RL ($D_0\approx0.8$) | $0.80_{\uparrow+.55}$ | +73.5% | $0.72_{\uparrow+.48}$ | +62.9% | $0.49_{\uparrow+.21}$ | +29.2% | $0.34_{\uparrow+.10}$ | +13.4% | $0.37_{\uparrow+.15}$ | +18.8% | +0.23 | +29.9% |
| SFT ($D_0\approx0.7$) | $0.69_{\uparrow+.43}$ | +58.1% | $0.44_{\uparrow+.20}$ | +26.3% | $0.29_{+.00}$ | +0.0% | $0.18_{\downarrow-.06}$ | -7.9% | $0.20_{\downarrow-.02}$ | -2.6% | +0.03 | +18.4% |
| RL ($D_0\approx0.7$) | $0.71_{\uparrow+.46}$ | +61.4% | $0.63_{\uparrow+.38}$ | +50.5% | $0.43_{\uparrow+.14}$ | +19.9% | $0.32_{\uparrow+.08}$ | +11.1% | $0.34_{\uparrow+.12}$ | +15.5% | +0.18 | +21.9% |
| SFT ($D_0\approx0.6$) | $0.61_{\uparrow+.35}$ | +47.4% | $0.41_{\uparrow+.17}$ | +21.9% | $0.31_{\uparrow+.02}$ | +3.0% | $0.19_{\downarrow-.05}$ | -7.1% | $0.22_{-.00}$ | -0.5% | +0.04 | +16.3% |
| RL ($D_0\approx0.6$) | $0.60_{\uparrow+.35}$ | +46.5% | $0.54_{\uparrow+.30}$ | +39.0% | $0.38_{\uparrow+.10}$ | +13.6% | $0.33_{\uparrow+.09}$ | +12.3% | $0.35_{\uparrow+.13}$ | +16.8% | +0.15 | +11.7% |
| SFT ($D_0\approx0.5$) | $0.51_{\uparrow+.26}$ | +34.7% | $0.40_{\uparrow+.16}$ | +20.8% | $0.32_{\uparrow+.04}$ | +5.2% | $0.21_{\downarrow-.03}$ | -4.3% | $0.26_{\uparrow+.04}$ | +4.5% | +0.05 | +12.9% |
| RL ($D_0\approx0.5$) | $0.49_{\uparrow+.23}$ | +31.2% | $0.49_{\uparrow+.25}$ | +33.0% | $0.34_{\uparrow+.06}$ | +7.8% | $0.34_{\uparrow+.10}$ | +13.6% | $0.31_{\uparrow+.09}$ | +11.3% | +0.12 | +8.0% |
| SFT ($D_0\approx0.4$) | $0.42_{\uparrow+.16}$ | +22.0% | $0.41_{\uparrow+.17}$ | +22.7% | $0.29_{+.00}$ | +0.3% | $0.19_{\downarrow-.05}$ | -6.2% | $0.19_{\downarrow-.03}$ | -4.4% | +0.02 | +16.8% |
| RL ($D_0\approx0.4$) | $0.41_{\uparrow+.16}$ | +21.0% | $0.34_{\uparrow+.09}$ | +12.4% | $0.32_{\uparrow+.03}$ | +4.8% | $0.24_{+.00}$ | +0.5% | $0.29_{\uparrow+.06}$ | +8.3% | +0.05 | +4.2% |

solution surface forms, preserving linguistic competence across implementation languages.

The second diagnostic focuses on $D_2$, where the learned solution family must survive recontextualization rather than implementation-language transfer. Unlike $D_1$, where the problem statement remains recognizable, $D_2$ changes the narrative while preserving the executable task. The model can no longer rely on remembered statement wording to recognize the seed problem. Figure 2b therefore compares one-to-one $D_0/D_2$ algorithm-mismatch failures, asking whether training improves structural solution-family identification rather than only reducing local implementation mistakes under a familiar surface form.

The $D_0$ rows provide the control: on the original seed problems, both trainable methods reduce algorithm-mismatch failures, from 116 for the base model to 44 for RL and 62 for SFT. Supervised imitation can therefore learn the appropriate solution family when the training and evaluation narratives are aligned. The contrast appears after $D_2$ recontextualization: the base model produces 177 algorithm-mismatch failures, RL reduces this count to 125, while SFT remains close to the base model at 172. The $D_2$ gap is therefore not explained by an inability of SFT to fit seed algorithms. Rather, SFT's algorithm-selection gain is largely tied to recognizing the original statement, whereas RL preserves more of that gain when the problem must be identified through structure. Appendix Table 7 reports the full taxonomy.

### 4.4. Sensitivity Analysis

The preceding analyses establish clear patterns (notably near-to-far attenuation, RL's positive transfer profile, and

ICL's strong-but-fragile profile), but rest on a single model trained on 64 seed problems of moderate difficulty, with ICL using oracle pairing. We vary the ICL retrieval method, model architecture, training set size, and problem difficulty to test robustness.

**ICL retrieval.** Paired ICL assumes oracle access to a relevant demonstration. We replace the oracle with three practical retrievers over the same 64-seed pool: text embedding (E5-Mistral cosine similarity), embedding + LLM rerank, and an LLM selector over all 64 candidate statements. Random retrieval serves as a lower control. The LLM selector recovers 100% recall on $D_0$–$D_2$ and nearly matches oracle ICL performance there, while embedding-only retrieval degrades as surface similarity decreases; at $D_3$–$D_4$, no method finds the structurally paired seed (Appendix F.2). ICL's near-transfer advantage thus survives dropping the oracle, while its far-transfer ceiling reflects a retrieval bottleneck (identifying algorithmic correspondence beyond surface similarity) rather than a contingent assumption.

**Model family.** We replicate the full protocol on DeepSeek-R1-Distill-Llama-8B, a Llama-based 8B model trained via distillation from DeepSeek-R1. Despite the change in architecture and scale, all principal findings transfer (Appendix G.3): transfer gains attenuate across the Spectrum, RL retains its advantage at matched $D_0$, and all methods converge at $D_3$–$D_4$.

**Seed set scale.** Next, we vary the number of seed problems. Expanding from 64 to 256 seeds preserves both the shape of the generalization profile and the relative ordering of methods (Appendix G.4). Under matched-memorization comparison (Figure 5), RL retains advantage over SFT on $D_1$, with attenuated but persistent gaps at $D_2$, while all

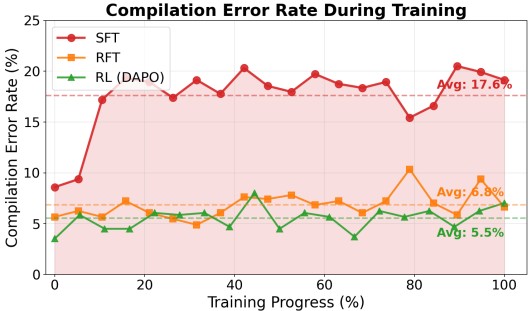

*(a)* $D_1$ compilation error rate during training.

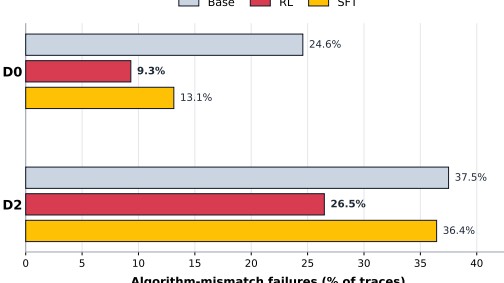

*(b)* Algorithm-mismatch failures on $D_0$ and $D_2$.

*Figure 2.* Near-transfer failure diagnostics. Top: SFT shows a persistently higher $D_1$ compilation-error rate than RL. Bottom: both SFT and RL reduce algorithm mismatch on $D_0$, but only RL preserves a substantial reduction after $D_2$ recontextualization.

methods converge at far-transfer ($D_3$–$D_4$).

**Hard problems.** The most demanding test examines problems where the base model achieves pass@32 = 0, creating conditions of sparse reward for RL and maximal distribution shift for SFT. Learning dynamics on this subset differ qualitatively: Gain(0) is severely depressed early in training as reward signals are rare, though sporadic correct solutions eventually accumulate. Nevertheless, conditioning on equivalent Gain(0) reveals the same structural patterns: transfer attenuates with distance, and the SFT–RL gap remains concentrated at near-transfer distances ($D_1/D_2$), with far-transfer improvement remaining bounded regardless of method (Figure 6). This suggests that the generalization radius is relatively robust to learning content.

# 5. Variants under the Spectrum: How Within-Family Modifications Deform Profiles

Having established the generalization profiles of vanilla ICL, SFT, and RL (§4), we next ask how stable these profiles are under within-family modifications. Many recent methods improve aggregate performance by enriching the learning signal, through reasoning demonstrations (Wei et al., 2022b; Zelikman et al., 2022), self-generated supervised tar-

gets (Yuan et al., 2023), self-distillation (Zhao et al., 2026; Shenfeld et al., 2026; Hübotter et al., 2026), or hint- and scaffold-assisted RL variants (Li et al., 2026; Liu et al., 2025b; Zhang et al., 2026; 2025; Wang et al., 2026; Huang et al., 2025a; Kang et al., 2025). The Generalization Spectrum lets us ask whether these variants expand the generalization radius of their base family, preserve it while changing profile height or speed, or compress it.

## 5.1. ICL Content Variants: Abstraction and Hints Lift Local Transfer

Generalization performance under paired ICL (Brown et al., 2020; Dong et al., 2024) depends on two steps: extracting a reusable algorithmic abstraction from the seed solution and recognizing when the target shares it. We vary demonstration content and correspondence hints to probe these steps, then combine them to test whether their effects overlap.

To probe extraction, we augment the raw-code demonstration with pseudocode that strips language-specific syntax (Code + Pseudocode), a short key insight (Code + Key Insight), or the natural reasoning trace that produced the code (Code + CoT) (Wei et al., 2022b; Zelikman et al., 2022). The Code Only baseline uses the raw solution alone. As Table 4 shows, pseudocode and key insight, which expose algorithmic structure explicitly, yield consistent gains on $D_0$–$D_2$, where the demonstration is instance-paired with the target. Code + CoT stays close to Code Only across these levels. Transfer therefore improves when the added content states the algorithm, not merely the derivation. At far-transfer distances ($D_3$, $D_4$), all three additions remain within run-to-run variation.

To probe recognition separately, we prepend a level-specific oracle hint that names the demonstration-target relation (e.g., "same algorithm, different language" at $D_1$; full hint texts in Appendix F.1). As Table 4 shows, hints yield little improvement at $D_0$, where correspondence inference is minimal, but produce substantial gains at $D_1$–$D_2$, where the model must infer correspondence across a language or narrative shift. At $D_3$ and $D_4$, where seed and target no longer share an instance-level pairing, hints no longer move pass@1.

Combining abstraction and hints yields additional gains on $D_1$–$D_2$ beyond either intervention alone ($D_2$: 64.4% → 80.1%), consistent with partially non-overlapping sources of error. Past $D_2$, all four conditions cluster around the code-only baseline, so neither richer content nor explicit correspondence cues extend transfer beyond the language- and narrative-shift regime.

The ICL interventions therefore move the paired portion of the Spectrum without shifting its boundary. The discontinuity lies between instance-level correspondence ($D_0$–$D_2$) and category-level pairing ($D_3$): $D_3$ behaves closer to un-

*Table 4.* Pass@1 under ICL content and correspondence-hint interventions. The Demonstration Format block reports Base, Code Only, Code + CoT, Code + Pseudocode, and Code + Key Insight. The Oracle Hint block reports variants where a level-specific cue naming the demonstration-target relation is prepended to raw code or to code with pseudocode (Appendix F.1). Values are percentages. Structured content and hints lift $D_0$–$D_2$ but stay near the code-only baseline at $D_3$–$D_4$.

| Level | Demonstration Format | | | | | Oracle Hint | |
|---|---|---|---|---|---|---|---|
| | Base | Code + CoT | Code Only | Code + Pseudocode | Code + Key Insight | +Hint | +Pseudocode + Hint |
| $D_0$ | 26.0% | 65.0% | 66.4% | 73.6% | 71.5% | 67.6% | 70.9% |
| $D_1$ | 24.0% | 57.0% | 56.6% | 63.3% | 63.9% | 62.9% | 68.0% |
| $D_2$ | 29.0% | 63.0% | 64.4% | 71.1% | 69.7% | 71.7% | **80.1%** |
| $D_3$ | 22.5% | 10.0% | 11.5% | 12.9% | 9.8% | 11.9% | 9.6% |
| $D_4$ | 19.0% | 10.0% | 11.3% | 11.3% | 12.7% | 14.1% | 11.3% |

paired $D_4$ than to local transfer.

## 5.2. SFT Target-Source Variants: RFT Preserves Far Transfer Relative to Demonstration SFT

With training problems and supervised objective held fixed, Figure 3 separates target-source alignment from the imitation loss. Demonstration SFT imitates the supervised demonstrations used as the reference baseline. RFT (Yuan et al., 2023) samples the base model and retains only rollouts that pass all tests, making it **SFT with self-generated correct targets**. GPT-OSS SFT imitates GPT-OSS-20B solutions and serves as a more distant off-policy source. GRPO is included only as a matched-memorization RL reference.

At matched memorization, target source mainly affects what survives beyond seed recall. RFT and demonstration SFT have nearly identical $D_0$ and $D_1$ performance, but RFT preserves a smoother tail from $D_2$ through $D_4$, whereas demonstration SFT falls after implementation transfer. GPT-OSS SFT drops earlier still, with weaker $D_1/D_2$ and the lowest far-transfer scores. The associated target-source perplexities follow the same ordering (1.30, 2.51, and 4.29), suggesting that larger source mismatch can support recall without carrying learned information into transfer.

This source effect is consistent with the $D_1$ failure-mode analysis in §4.3 and Figure 2a. Demonstration SFT raises the average compile-error rate on C++ transfer to 17.6%, whereas RFT with self-generated targets remains low at 6.8%. The diagnostic suggests that the problem is not only weaker semantic transfer; imitating external Python traces can also interfere with implementation-language competence. RL's 5.5% compile-error rate is reported in §4.3 as part of the algorithm comparison, while the SFT-family contrast here is between demonstration SFT and RFT. This pattern points to target-source alignment, rather than the supervised objective alone, as the factor that determines whether imitation preserves or compresses far transfer.

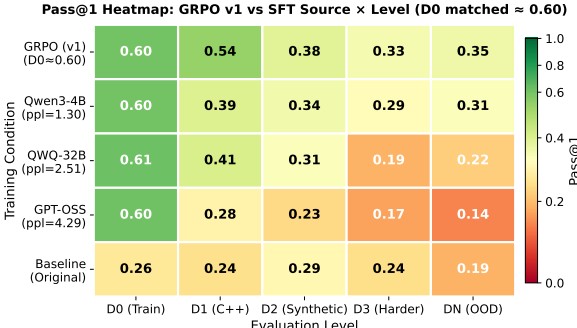

*Figure 3.* SFT target-source variants at matched $D_0 \approx 0.6$. The direct SFT-family comparison is RFT with self-generated Qwen3-4B targets against demonstration SFT with QWQ-32B targets. GPT-OSS-20B provides a more distant off-policy extrapolation, and GRPO is included as the matched-memorization RL reference.

## 5.3. Self-Distillation Variants: An Implicit Demonstration Reward Lifts Local Transfer but Weakens Far Transfer

Unlike the target-source variants, SDFT changes the role of the demonstration itself. Rather than being copied as a target sequence, the demonstration shapes a dense preference signal over the model's own rollouts (Shenfeld et al., 2026; Zhao et al., 2026; Hübotter et al., 2026). For samples from $\pi_\theta(\cdot|x)$, the objective applies a reverse-KL loss against the demonstration-conditioned policy $\pi_T(\cdot|x, c)$ (Shenfeld et al., 2026). The matched-memorization protocol isolates how this inverse-RL-like signal changes the Spectrum profile.

At matched $D_0 \approx 0.6$, SDFT produces a local-transfer gain with a far-transfer cost (Figure 4; full numbers in Appendix G.2). It outperforms vanilla SFT on $D_1$ and $D_2$, where the evaluation item remains an implementation or narrative variant of the paired seed. At $D_3$ and $D_4$, however, it falls below both vanilla SFT and the unadapted base model, indicating that the gain is confined to near correspondence rather than extending to category-paired or unpaired transfer.

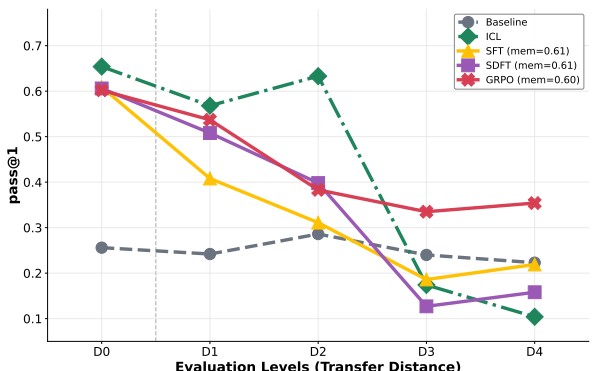

*Figure 4.* SDFT vs. vanilla SFT at matched $D_0 \approx 0.6$. SDFT lifts $D_1/D_2$ above SFT, but its far-transfer profile drops below SFT at $D_3/D_4$. Full numeric profile in Appendix G.2.

Viewed through the profile rather than the objective alone, SDFT behaves as an **ICL-distillate**: it writes demonstration-conditioned local structure into the weights, but does not preserve that benefit once the target no longer shares an instance-level correspondence.

### 5.4. RL Algorithmic Variants: Training-Time Hints Improve Sample Efficiency but Compress Far Transfer

Recent hint- and scaffold-assisted RL methods seek to reduce the exploration burden of sparse verifier rewards by adding partial solutions, expert anchors, stepwise hints, heuristic guidance, or hint-completion pairs during training (Li et al., 2026; Liu et al., 2025b; Zhang et al., 2026; 2025; Wang et al., 2026; Huang et al., 2025a; Kang et al., 2025). In this setting, the scaffold is expected to help RL make more efficient use of each seed sample. What remains unclear is whether this more efficient use of the training sample also produces a broader generalization profile after learning. To test this, we compare verifier-only GRPO (Shao et al., 2024) with two **hint-assisted GRPO** variants at matched memorization ($D_0 \approx 0.6$). Hint GRPO-Pseudo appends pseudocode to the training sample, while Hint GRPO-KI names the key algorithmic idea. Both variants provide compact algorithmic scaffolds rather than full target solutions.

The training trajectory confirms the intended sample-efficiency effect. Both hint variants reach the same memorization level much earlier than verifier-only GRPO, with Hint GRPO-KI at step 60, Hint GRPO-Pseudo at step 70, and GRPO at step 170 (Figure 7). Yet this faster adaptation does not translate into broader generalization. At $D_0 \approx 0.6$, hint-assisted GRPO stays close to verifier-only GRPO on $D_1$ and $D_2$, indicating that the scaffold preserves local implementation and narrative transfer under matched memorization. The separation appears at the far end of the Spectrum, where both hint variants fall below verifier-

only GRPO on $D_3$ and $D_4$ (Figure 8; exact values in Appendix G.2). Training-time hints therefore improve the speed of fitting the seed examples, but do not expand the generalization radius. A plausible interpretation is that the scaffold makes reward-bearing trajectories easier to find while keeping learning tied to the scaffolded seed relation. Under the Spectrum, this appears as faster memorization with a narrower far-transfer tail.

## 6. Conclusions

We introduce the **Generalization Spectrum**, a chromatographic-style evaluation that measures not only *whether* a learning algorithm improves, but **how far** improvements transfer from each training example across controlled distances (D0–D4). By instantiating this framework in competitive programming and releasing a paired benchmark with **256 evaluation instances**, we enable fine-grained diagnosis of memorization, near-transfer, and far-transfer behavior under a unified protocol.

Across in-context learning, supervised fine-tuning, and reinforcement learning, we find a consistent pattern: generalization decays with transfer distance, though the shape varies by paradigm. Paired ICL's transfer is largely confined to instance-level correspondence ($D_0$–$D_2$), whereas under matched memorization, RL converts memorization into near-transfer more efficiently than SFT-family baselines and preserves a more positive far-transfer tail. A finer-grained comparison of recent within-family variants further sharpens this picture. Abstraction and hints lift paired ICL without extending $D_3/D_4$, on-policy or self-generated supervised targets preserve transfer better than off-policy imitation, and SDFT and hint-assisted GRPO improve local transfer or optimization speed but can compress the far-transfer tail. Across variants, auxiliary signals do not simply improve generalization; they move different parts of the Spectrum, separating local gains, far-transfer preservation, and optimization speed.

**Limitations and Future Work.** Our study uses competitive programming, a domain chosen for its clear evaluation and structured variation. We believe the Generalization Spectrum framework is domain-agnostic; its validity across mathematical reasoning, code translation, and creative generation remains for future work. Looking ahead, the Spectrum suggests concrete directions: increasing the number of variants per seed to better resolve decay profiles, exploring mixed training pipelines that combine complementary profiles, such as GRPO followed by distillation, and developing training methods that explicitly optimize **generalization efficiency** rather than memorization alone. We hope this framework enables more nuanced evaluation and inspires algorithms that learn not only patterns, but how to adapt them meaningfully to new challenges.

## Impact Statement

This paper presents work whose goal is to advance the field of machine learning. There are many potential societal consequences of our work, none of which we feel must be specifically highlighted here.

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

## A. Related Work

**Generalization of Post-Training Paradigms.** A growing body of work compares how ICL, SFT, and RL generalize under distribution shift, building on the observation that large language models acquire scale-dependent emergent abilities (Wei et al., 2022a). For ICL versus fine-tuning, evidence is mixed: ICL can outperform fine-tuning on tasks with implicit regularities via context-induced circuit shifts (Yin et al., 2024), while fine-tuning achieves stronger performance given sufficient data (Liu et al., 2022). Many-shot ICL can match fine-tuning in some regimes (Agarwal et al., 2024), though ICL's generalization appears bounded by the pretraining hypothesis space (Wang et al., 2024; Goddard et al., 2025). For SFT versus RL, comparisons more consistently report improved robustness for RL as distribution shift increases (Kirk et al., 2024; Chu et al., 2025; Huan et al., 2025; Luong et al., 2024). Beyond these core paradigms, recent variants modify the learning signal: rejection-sampling fine-tuning (RFT) (Yuan et al., 2023) uses self-generated on-policy targets, hint- and scaffold-assisted RL methods provide partial solutions, expert anchors, stepwise hints, heuristic guidance, or hint-completion pairs to mitigate sparse rewards (Li et al., 2026; Liu et al., 2025b; Zhang et al., 2026; 2025; Wang et al., 2026; Huang et al., 2025a; Kang et al., 2025), and self-taught distillation methods (Shenfeld et al., 2026; Zhao et al., 2026) bridge context-based and parameter-based learning. However, all these studies assess generalization through aggregate metrics on coarse in-distribution versus out-of-distribution splits, making it difficult to quantify *how far* transfer extends from individual training examples.

**Beyond Binary Generalization Evaluation.** Standard OOD evaluation treats generalization as binary—in-distribution or out—obscuring the gradual decay of transfer with increasing distance (Ye et al., 2022; Yu et al., 2024). In reasoning domains, ThinkBench (Huang et al., 2025b) and related efforts construct OOD variants dynamically, yet still report binary ID/OOD comparisons. Our work differs by constructing a *paired* evaluation spectrum: each training example maps to test variants at multiple controlled distances (D0–D4), enabling measurement of per-sample generalization profiles rather than aggregate OOD gaps.

**Benchmarks and Synthetic Data for Code Reasoning.** Competitive programming offers unique advantages for studying generalization: correctness is unambiguous via test cases (Chen et al., 2021; Austin et al., 2021), problems encode implicit algorithmic patterns resistant to surface memorization, and continuous new releases mitigate contamination (Jain et al., 2024; Quan et al., 2025; Zheng et al., 2025). Prior work on synthetic coding data spans problem–solution–test synthesis (Liu et al., 2025a; Xu et al., 2025; Zeng et al., 2025), automatic test-case generation and specification validation (Gabel & Su, 2012; He et al., 2025; Ma et al., 2025; Wang et al., 2025; Fu et al., 2025), and reasoning trace or self-critique construction (Ahmad et al., 2025b;a). Cross-lingual benchmarks evaluate implementation transfer across languages (Cassano et al., 2023; Peng et al., 2024), while controlled generators and template-based methods probe transfer across algorithmic variants (Sun et al., 2025; Zhou et al., 2025; Tang et al., 2025). We build on these directions by constructing paired variants at controlled transfer distances—from cross-language transfer (D1) through narrative reframing (D2) to category-matched problems (D3)—enabling fine-grained diagnosis of where each learning paradigm's generalization breaks down.

## B. Supplementary Robustness Figures

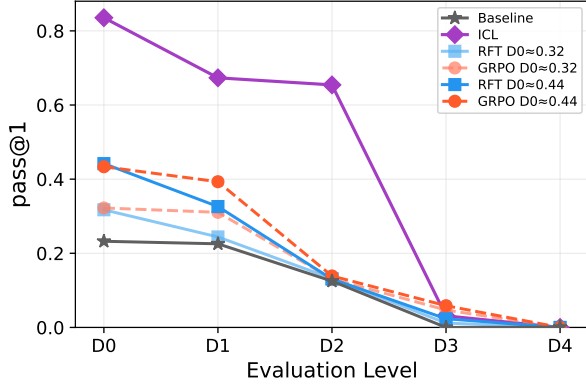

*Figure 5.* Pass@1 across the Generalization Spectrum ($D_0 - D_4$) with 256 seeds. GRPO outperforms SFT on near-transfer ($D_1 - D_2$); all methods converge at far-transfer.

*Figure 6.* Generalization profiles on the hard subset (pass@32 = 0) under matched Gain(0). The distance-decay pattern persists; RL retains its advantage at near distances ($D_1/D_2$).

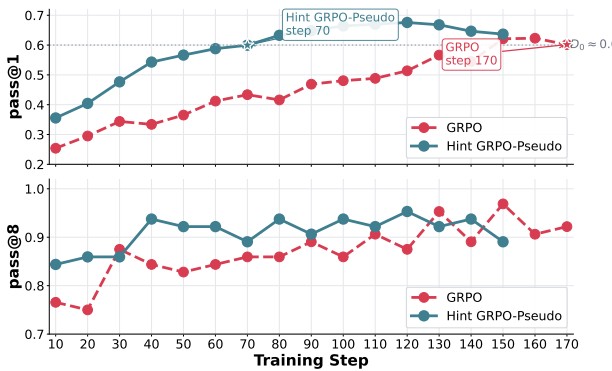

*Figure 7.* Matched-checkpoint steps at $D_0 \approx 0.6$: Hint GRPO-Pseudo at step 70 and Hint GRPO-KI at step 60, versus GRPO at step 170.

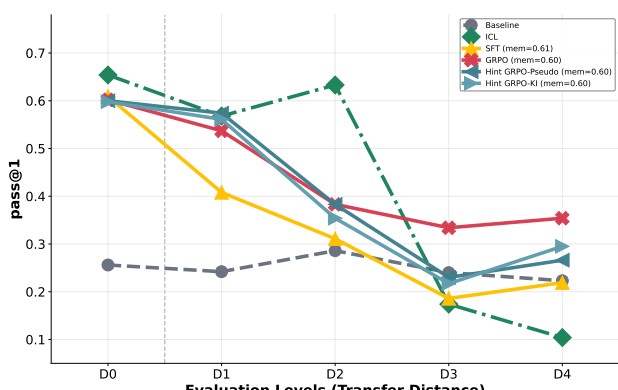

*Figure 8.* Under matched $D_0$, Hint GRPO stays close to GRPO at $D_1/D_2$ but falls below it at $D_3/D_4$. Full numeric profile in Appendix G.2.

## C. Dataset Construction Details

This section provides the full details of our dataset construction pipeline, including selection criteria, synthesis procedures, and quality control measures.

### C.1. Why Competitive Programming

Instantiating the Generalization Spectrum framework requires a domain with precise correctness criteria and rich structure for controlled variations. We use competitive programming as our testbed for the following reasons:

- **No room for lucky guesses.** Unlike multiple-choice or short-answer questions, the space of correct solutions is vanishingly sparse—a random string has essentially zero probability of passing all test cases. Success is therefore unlikely to arise from statistical flukes alone.

- **Unambiguous correctness.** Each problem comes with test cases $\{(x_i, y_i)\}$; correctness is determined by execution, requiring no human judgment.

- **Rich space for variations.** A single algorithmic idea naturally instantiates across different problem statements, constraints, and application contexts—providing an ideal basis for constructing varying transfer distances.

- **Ever-growing problem pool.** Platforms like Codeforces and AtCoder continually release new problems, supplying fresh test material and reducing contamination risk. This lets the Generalization Spectrum evolve alongside model

capabilities.

## C.2. Construction Pipeline

We construct five problem sets (D0–D4) via two strategies: **selection** from existing problems (D0, D1, D3, D4) and **synthesis** of new problems (D2). We detail each below.

**D0 (Seed Problems).**  We aggregate Codeforces problems from recent competitive programming datasets: RStar-Coder (Liu et al., 2025a), CodeContests+ (Wang et al., 2025), and LiveCodeBench-Pro (Zheng et al., 2025).

We first filter by **difficulty**: problems where the base model (Qwen3-4B-Thinking) achieves an average test case pass rate below 0.6. This ensures sufficient challenge—trivially easy problems cannot differentiate learning algorithms. We then split by **learnability**: whether the model can solve the problem within a few attempts. Main experiments focus on learnable problems (pass@8 $> 0$); harder problems (pass@32 $= 0$) are analyzed separately in §4.4. Within qualifying problems, we **sample randomly** with no constraints on algorithmic category.

This yields 64 seed problems. Appendix G.4 describes the corresponding seed-scale ablation with 256 problems.

**D1 (Implementation Transfer).**  We reuse D0 problems directly, changing only the target language from Python to C++ at evaluation time. No additional construction needed.

**D2 (Context Transfer).**  D2 requires generating problems with entirely different narrative contexts—not paraphrases, but complete recontextualizations where characters, settings, and stories change while the underlying mathematical structure remains identical. The solution and test cases are preserved. We use a four-stage pipeline:

1. **Solution verification.** GPT-5.2 attempts the original problem (pass@3). Only problems it solves proceed—this ensures the LLM understands the problem well enough to generate valid variants.

2. **Problem generation.** Given the original statement, I/O format, and examples, we prompt GPT-5.2 to generate a new problem description. The prompt requires: (a) entirely different story and narrative, (b) identical I/O format, (c) the original solution logic must still apply.

3. **Consistency review.** Gemini-3-Pro independently checks whether the new problem truly shares the same solution and test cases. Failed cases are filtered out or regenerated. First-pass acceptance is approximately 85%; after regeneration, retention is 100%.

4. **Solution re-derivation.** As final validation, Gemini-3-Pro solves the new problem from scratch (no access to the original) and we verify that its solution passes the original test cases. This provides further evidence of semantic equivalence.

For quality control, we automatically verify that the original solution still passes all test cases on the new problem, and manually spot-check 10% for natural phrasing and unambiguous specification. The Codeforces-style ratings reported for D2 are GPT-5 estimates calibrated to the original rating scale and are used only as descriptive metadata; D2 selection inherits the D0 difficulty and learnability filter.

**D3 (Category Matched).**  We select problems sharing algorithmic tags with seed problems, testing whether learning one algorithm (e.g., BFS) transfers to different problems requiring the same algorithmic skill. Candidate problems satisfy the same difficulty and learnability filter as D0: base-model solve rate below 0.6 and pass@8 $> 0$.

*Classification system.* We use a hierarchical taxonomy of 30 algorithm categories organized into six families: **DS** (Data Structures: arrays, hash tables, heaps, stacks, segment trees), **DP** (Dynamic Programming: classic, digit, interval, tree DP), **GT** (Graph Theory: traversal, shortest path, MST, flow), **M** (Mathematics: number theory, combinatorics, geometry), **SP** (String Processing), and **Others** (greedy, binary search, two pointers, simulation, etc.). Each problem is assigned 1–3 categories by GPT-based classification using both the problem statement and solution code.

*Tag source.* The tag source depends on problem difficulty: for rating $\leq 2000$, we use LLM annotations (official Codeforces tags are crowd-sourced and noisy at lower difficulties); for rating $> 2000$, we use official Codeforces tags (curated by expert users and more reliable than LLM annotations for hard problems).

*Matching algorithm.* We employ **family-based matching**: categories are first mapped to their family prefix (e.g., "DP – Classic DP" → "DP") before computing overlap, capturing algorithmic similarity at an appropriate granularity. We then apply **maximum bipartite matching** (Hungarian algorithm) to find optimal 1-to-1 pairings between D0 and the candidate pool, ensuring no problem is reused and maximizing total matches. This achieves 100% matching success (64/64 pairs). Among matched pairs, 92.2% share exactly one family tag and 7.8% share two tags. The most common matching families are Data Structures (15 pairs), Greedy (14), Mathematics (13), and Dynamic Programming (4).

**D4 (Unpaired Baseline).** We use random samples from the same source and time period as seed problems, excluding those already used in D0 or D3, and applying the same difficulty and learnability filter as D0. This represents in-domain generalization with no direct relation—measuring the marginal contribution of learning a single sample to overall domain ability.

### C.3. Quality Control Summary

Table 5 summarizes the quality control measures applied at each level.

*Table 5.* Quality control measures for each spectrum level.

| Level | Verification Method | Pass Criteria |
|-------|---------------------|---------------|
| D0 | Difficulty filter + learnability check | pass rate $< 0.6$, pass@8 $> 0$ |
| D1 | Automatic (same problem) | N/A |
| D2 | Multi-model generation + cross-validation | Solution equivalence verified |
| D3 | Tag matching + bipartite matching | $\geq 1$ shared family tag |
| D4 | Random sampling + exclusion filter | Not in D0/D3 |

The cosine similarity reported in Table 2 is computed between the sentence embeddings of each variant's problem statement and its paired D0 statement, using the all-mpnet-base-v2 model (Reimers & Gurevych, 2019).

## D. Example Test Variants

We illustrate the Generalization Spectrum with concrete examples from our dataset. The following shows a seed problem $(D_0)$, its recontextualized variant $(D_2)$, a category-matched problem $(D_3)$, and an unpaired baseline problem $(D_4)$.

### $D_0$: Seed Problem (Grid Reachability)

---

**Original Problem**

NEKO has just got a new maze game on her PC!

The game's main puzzle is a maze, in the form of a $2 \times n$ rectangle grid. NEKO's task is to lead a Nekomimi girl from cell $(1, 1)$ to the gate at $(2, n)$ and escape the maze. The girl can only move between cells sharing a common side.

However, at some moments during the game, some cells may change their state: either from normal ground to lava (which forbids movement into that cell), or vice versa. Initially all cells are of the ground type.

After hours of streaming, NEKO finally figured out there are only $q$ such moments: the $i$-th moment toggles the state of cell $(r_i, c_i)$. Knowing this, NEKO wonders, after each of the $q$ moments, whether it is still possible to move from cell $(1, 1)$ to cell $(2, n)$ without going through any lava cells.

**Input:** $n, q$ ($2 \leq n \leq 10^5$, $1 \leq q \leq 10^5$), followed by $q$ lines of $(r_i, c_i)$.
**Output:** For each moment, print "Yes" or "No".

---

### $D_2$: Synthetic Variant (Same Algorithm, Different Context)

---

**Synthetic Problem**

Two parallel railway lines run side-by-side, each divided into $n$ numbered segments from west to east. A maintenance trolley starts on the northern line at segment 1 and must reach the depot on the southern line at segment $n$. The trolley can move between segments that share a common side: it may move east or west along the same line, or switch between the two lines within the same segment index (i.e., vertical move).

---

Some segments may become temporarily closed for maintenance, and later reopen. Initially, all segments are open. You are given $q$ events; the $i$-th event toggles the status of segment $(r_i, c_i)$: if it was open it becomes closed, and if it was closed it becomes open. After each event, determine whether the trolley can still reach the depot from its start without entering any closed segment.

**Input:** $n$ and $q$ ($2 \leq n \leq 10^5$, $1 \leq q \leq 10^5$), followed by $q$ lines of $(r_i, c_i)$.
**Output:** After each event, print "Yes" or "No".

The $D_2$ variant preserves the exact algorithmic structure (dynamic connectivity on a $2 \times n$ grid with toggle operations) while completely changing the surface narrative from a video game maze to a railway maintenance scenario. The input/output format and constraints are identical.

### $D_3$: Category Matched (Same Algorithm Family)

**Category-Matched Problem**

In the NN country, there are $n$ cities, numbered from 1 to $n$, and $n - 1$ roads connecting them. There is a road path between any two cities.
There are $m$ bidirectional bus routes between cities. Buses drive between two cities taking the shortest path with stops in every city they drive through. Travelling by bus, you can travel from any stop on the route to any other. You can travel between cities only by bus.
You are interested in $q$ questions: is it possible to get from one city to another and what is the minimum number of buses you need to use for it?

**Input:** $n$ ($2 \leq n \leq 2 \cdot 10^5$), $n - 1$ parent pointers, $m$ bus routes, $q$ queries.
**Output:** For each query, print the minimum number of buses or $-1$ if unreachable.

The $D_3$ problem shares the algorithmic category (graph traversal / shortest path) with $D_0$ but requires different techniques (tree structure, BFS on route graph). This tests whether learning one graph algorithm transfers to related problems in the same family.

### $D_4$: Unpaired Baseline

**Unpaired Baseline Problem**

Gildong was hiking a mountain, walking by millions of trees. Inspired by them, he suddenly came up with an interesting idea for trees in data structures: What if we add another edge in a tree?
Then he found that such tree-like graphs are called 1-trees. First, he'll provide you a tree with $n$ vertices, then he will ask you $q$ queries. Each query contains 5 integers: $x$, $y$, $a$, $b$, and $k$. This means you're asked to determine if there exists a path from vertex $a$ to $b$ that contains exactly $k$ edges after adding a bidirectional edge between vertices $x$ and $y$. A path can contain the same vertices and same edges multiple times. All queries are independent of each other.

**Input:** $n$ ($3 \leq n \leq 10^5$), $n - 1$ edges, $q$ queries ($1 \leq q \leq 10^5$), each with $(x, y, a, b, k)$.
**Output:** For each query, print "YES" or "NO".

The $D_4$ problem is randomly sampled from the same time period as D0 with no direct pairing, serving as a reference for in-domain generalization. It tests whether training on $D_0$ provides any general domain benefit beyond sample-specific transfer.

## E. Experimental Setup

### E.1. Training Configuration

**Learning paradigms.** All paradigms learn from the same 64 seed problems, each paired with a reference solution containing the problem statement, step-by-step reasoning trace, and Python code. For ICL, we prepend a single paired seed problem with its reference solution as a 1-shot demonstration. The paired setting uses the oracle pair mapping: each evaluation problem is conditioned on its corresponding matched seed problem. The random baseline samples the demonstration without using this pair mapping. No model parameters are updated for ICL.

SFT fine-tunes the base model on reference (problem, reasoning trace, code) triples. RFT uses the same training objective and hyperparameters as SFT, but replaces reference solutions with self-generated correct solutions: the base model samples

candidate solutions, and we retain only those passing all test cases. RL uses outcome-based binary reward ($+1$ for passing all test cases, 0 otherwise). The main text reports GRPO as the primary RL algorithm; extended GRPO and DAPO profiles are reported in Table 12.

**Matched-memorization selection.** Different learning paradigms memorize seed problems at different rates. A method with higher $D_0$ accuracy has more opportunity to exhibit transfer, so comparing final checkpoints would mix two factors: how much the method has memorized and how efficiently that memorization transfers. We therefore save checkpoints at regular intervals, evaluate each checkpoint on $D_0$ (exact recall of seed problems), and compare methods at matched $D_0$ pass@1 levels such as 50%, 60%, 70%, and 80%. ICL has no intermediate checkpoints, so its $D_0$ level is determined by the demonstration strategy.

Table 6 summarizes key hyperparameters for each gradient-based learning paradigm.

*Table 6.* Key hyperparameters for SFT, RFT, and RL training. Checkpoints are saved periodically to enable matched-memorization comparisons at equivalent $D_0$ performance levels.

|  | SFT | RFT | RL |
| --- | --- | --- | --- |
| Learning rate | 2e-5 | 2e-5 | 1e-6 |
| Batch size | 64 | 64 | 64 |
| Training steps | 1000 | 1000 | 150 |
| Warmup ratio | 0.1 | 0.1 | 0.05 |
| Rollouts per problem | — | — | 8 |
| KL coefficient | — | — | 0.01 |

For RL, 150 steps is the standard training budget. Selected runs were continued up to 400 steps for extended matched-memorization profiles, and Table 12 reports the actual checkpoint step used for each row.

### E.2. Implementation Details

**Evaluation protocol.** Each selected checkpoint is frozen and evaluated on all five spectrum levels ($D_0$–$D_4$), producing $64 \times 5 = 320$ total instances, of which $64 \times 4 = 256$ are transfer instances at $D_1$–$D_4$. We report **pass@1** as the primary metric. Each configuration is evaluated over **8 independent runs** with different random seeds, and we report the mean. Transfer metrics ($\text{Gain}(i)$, $\text{Gain}_n(i)$, AUS, N-F$_n$) follow the definitions in §2.3; matched $D_0$ comparison makes these profiles reflect transfer efficiency rather than unequal training progress.

**Decoding parameters.** We use Qwen3-4B-Thinking as the base model with thinking mode enabled (`enable_thinking=True`). For all methods, we sample with temperature 0.8, top-p 0.95, and maximum output length of 32768 tokens. Each evaluation run uses a unique random seed. Generation terminates upon producing the end-of-code delimiter or reaching the token limit.

**Execution environment.** Solutions are executed in an isolated sandbox with the following configuration:

- Python 3.10, GCC 11.4 (for C++17)

- Time limit: 5 seconds per test case

- Memory limit: 512 MB

- Solutions that fail to compile, exceed time/memory limits, or crash at runtime are marked as incorrect

**Error classification.** We classify failures into three categories for fine-grained analysis:

1. *Syntax/compile errors*—code fails to compile or parse;

2. *Runtime errors*—code compiles but crashes during execution;

3. *Wrong answer*—code runs to completion but produces incorrect output.

All categories count as fail for pass@1.

$D_0/D_2$ **failure taxonomy.** The $D_0/D_2$ diagnostic separates evaluator-failed traces by the first diagnosed cause of failure. The annotation was performed by GPT-5.2 Thinking, which was given the problem statement, the reference solution, the generated code, and the evaluator pass rate. The taxonomy is designed to distinguish failures in solution-family identification from closer specification or implementation errors:

1. *Algorithm mismatch*: the solution chooses an incompatible algorithmic route or solution family.
2. *Constraint omission*: the algorithmic direction is close, but a required constraint, boundary case, or output condition is missing.
3. *Structure confusion*: the broad family is similar, but the state space, transition, counted object, graph relation, or interval semantics is mis-specified.
4. *Other error*: the high-level route is plausible, but the code fails through implementation, indexing, I/O, variable, or complexity errors.

When the annotator judged an evaluator-failed trace to be apparently correct or could not assign a category, that trace is excluded from the failure-category counts. Table 7 reports counts for the aligned $D_0/D_2$ diagnostic set.

*Table 7.* Failure-taxonomy counts for the aligned $D_0/D_2$ diagnostic. Each row uses evaluator-failed traces from the one-to-one $D_0/D_2$ comparison, excluding traces marked as apparently correct or unknown by the annotator. Counts are shown without normalization.

| Level | Method | Algorithm mismatch | Constraint omission | Structure confusion | Other error |
|-------|--------|--------------------|--------------------|--------------------|-------------|
| $D_0$ | Base | 116 | 52 | 90 | 70 |
| $D_0$ | RL | 44 | 20 | 35 | 79 |
| $D_0$ | SFT | 62 | 30 | 37 | 60 |
| $D_2$ | Base | 177 | 46 | 51 | 56 |
| $D_2$ | RL | 125 | 34 | 45 | 62 |
| $D_2$ | SFT | 172 | 30 | 44 | 67 |

**Code extraction.** For models that generate chain-of-thought reasoning, only the final code block is extracted and executed; intermediate reasoning is ignored for evaluation.

**Prompt template.** All prompts follow a unified structure: language-specific instructions, problem description, and input/output specification. Training and evaluation use the same template, differing only in the specified target language (e.g., Python for training vs. C++17 for $D_1$ evaluation). The full prompt template is shown below:

```
Prompt Template

You are an expert competitive programmer.  You will be given a competitive
programming problem.  Please reason step by step about the solution, then provide a
complete implementation in {Python 3 | C++17}.
Your solution must read input from standard input, write output to standard output.
Do not include any debug prints or additional output.  Put your final solution within
a single code block:  ```{python|cpp} ...```

# Problem:  {title}
{problem_statement}
Execution time limit:  {time_limit} seconds
Memory limit:  {memory_limit} MB
## Input Format
{input_format}
## Output Format
{output_format}
## Examples
```input
{example_input}
```

```output
{example_output}
```

## Note
```

```
{note}
```

For ICL, we construct a multi-turn conversation with the demonstration as a completed interaction:

**ICL Format**

```
[
{"role":  "user",      "content":  "{example_prompt}"},
{"role":  "assistant", "content":  "{example_answer}"},
{"role":  "user",      "content":  "{target_prompt}"}
]
```

where `example_prompt` is the seed problem (same template as above), `example_answer` contains the chain-of-thought reasoning followed by the solution code, and `target_prompt` is the test problem to be solved.

**Model selection.** When multiple algorithmic variants are considered, we report their training-set $D_0$ improvement speed, measured by steps required to reach a fixed pass@1 threshold. SFT regularization variants are reported in Appendix G.1; full matched-memorization profiles for trainable variants are reported in Appendix G.2.

### E.3. Training-Run Generalization Profiles

Figure 9 reports the full checkpoint trajectories behind the matched-memorization comparisons. These curves serve a different role from Table 12: the table aligns methods at comparable $D_0$ levels, while the figure shows how each method reaches those levels during training. Across runs, $D_0$ usually improves fastest. The transfer levels then separate by method: $D_1$ and $D_2$ often rise with $D_0$, but with smaller gains, while $D_3$ and $D_4$ are flatter or more volatile. This supports using matched-$D_0$ comparisons, since raw final checkpoints would conflate transfer efficiency with different optimization speeds.

## F. Analysis Details

### F.1. ICL: Oracle Hint Texts

Table 8 lists the oracle hints prepended to the demonstration in the recognition-bottleneck experiment (§5.1). Each hint names the demonstration-target relation at the corresponding spectrum level.

*Table 8.* Oracle hint texts used in the recognition-bottleneck probing experiment.

| Level | Hint text |
|-------|-----------|
| $D_0$ | "Note: This is the exact same problem as the previous one." |
| $D_1$ | "This problem requires the same algorithm as the previous one, implemented in a different programming language." |
| $D_2$ | "This problem has identical solution logic as the previous one—only the problem description differs." |
| $D_3$ | "This problem uses a similar algorithmic approach as the previous one." |
| $D_4$ | "This is an unrelated problem from the same domain." |

### F.2. ICL: Realistic Retrieval

The main ICL experiment uses paired demonstrations, which isolate the effect of a relevant example but also represent an oracle retrieval setting. To test whether this assumption is attainable in the few-shot regime, we run a retrieval study using the same 64-seed training pool. Each method selects one seed demonstration for a target problem before applying the standard ICL prompt.

**Retrieval methods.** We compare five selection strategies: (i) Oracle, which uses the known paired seed; (ii) Text Embedding, which embeds problem statements with E5-Mistral-7B-Instruct and retrieves by cosine similarity; (iii) Embed + LLM Rerank, which reranks the top-20 embedding candidates with GPT-5.4-thinking; (iv) LLM Selector, which sees all 64 full seed statements in one context window and chooses the most algorithmically relevant demonstration; and (v) Random,

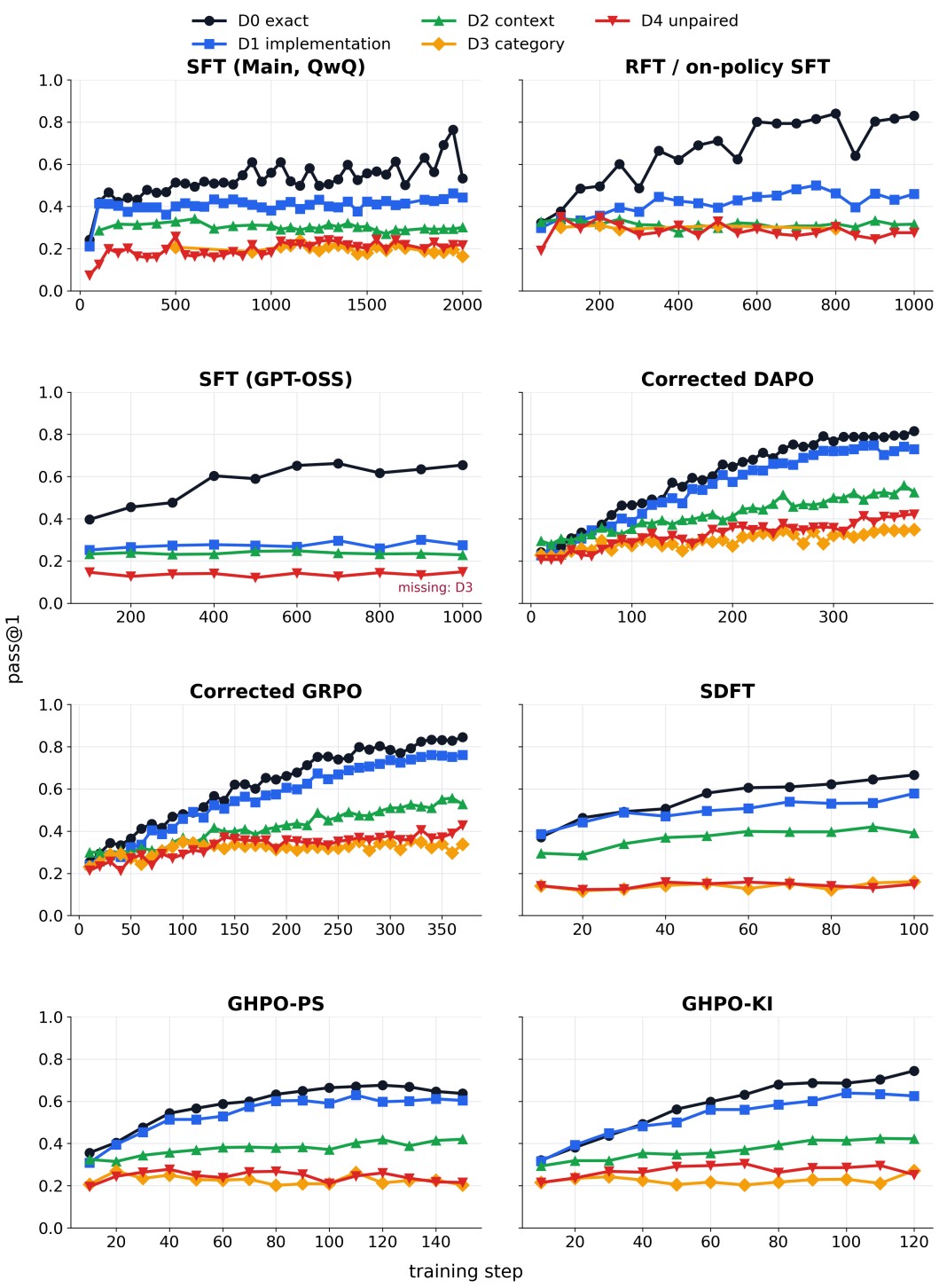

*Figure 9.* Full training-run generalization profiles. Each panel plots pass@1 across $D_0$–$D_4$ over saved checkpoints for one training method. The trajectories show the raw checkpoint behaviour used to select matched-memorization comparison points.

which samples a seed uniformly.

*Table 9.* Retrieval accuracy for selecting the true paired seed from the 64-seed pool. $D_0$ and $D_1$ share the same statement, so they have identical retrieval accuracy.

| Retrieval Method | $D_0/D_1$ | $D_2$ | $D_3$ |
|---|---|---|---|
| Oracle | 100% | 100% | 100% |
| LLM Selector | 100% | 100% | 0% |
| Embed + LLM Rerank | 100% | 78.0% | 0% |
| Text Embedding | 100% | 47.5% | 0% |
| Random | 1.6% | 1.6% | 1.6% |

*Table 10.* ICL pass@1 under different retrieval strategies. LLM selection nearly matches oracle performance on $D_0$–$D_2$, while all non-oracle methods provide limited gains on $D_3$–$D_4$, where the correct correspondence is algorithmic rather than statement-level.

| Retrieval Method | $D_0$ | $D_1$ | $D_2$ | $D_3$ | $D_4$ |
|---|---|---|---|---|---|
| Base model | 0.26 | 0.24 | 0.29 | 0.24 | 0.22 |
| Oracle | 0.65 | 0.57 | 0.63 | 0.17 | 0.10 |
| LLM Selector | 0.65 | 0.57 | 0.64 | 0.12 | 0.16 |
| Embed + LLM Rerank | 0.64 | 0.57 | 0.55 | 0.13 | 0.17 |
| Text Embedding | 0.65 | 0.58 | 0.44 | 0.15 | 0.20 |
| Random | 0.21 | 0.17 | 0.21 | 0.06 | 0.10 |

**Full-statement context is what makes the LLM selector work.** The LLM selector reaches 100% recall on $D_0$–$D_2$ when given the full $\sim$28k tokens of all 64 candidate statements in one context window. To check whether this success comes from the LLM's reasoning or merely from keyword matching, we truncate each candidate to its first 240 characters and rerun selection: $D_0$ recall collapses from 100% to 7.8%. The selector therefore relies on detailed problem semantics rather than surface keywords, which is why a text-embedding retriever over the same statements cannot match it on $D_2$.

**ICL performance tracks retrieval accuracy.** The pass@1 in Table 10 closely mirrors the recall in Table 9. On $D_2$, pass@1 drops from 0.64 (LLM Selector, 100% recall) to 0.55 (Embed + LLM Rerank, 78% recall) to 0.44 (Text Embedding, 47.5% recall), matching the recall ordering. On $D_3$, the practical non-random retrievers recover 0% of the paired seeds, while Random reaches 1.6% by chance; $D_4$ has no true paired seed. The corresponding pass@1 values remain below or close to the base model, indicating limited ICL gains beyond near transfer. This coupling is consistent with the paper's main claim: ICL transfer is gated by demonstration-test correspondence, and retrieval failure passes through to the downstream ICL result.

**Summary.** The key takeaway is not the absolute 100% recall on a small candidate pool, but the qualitative gap between correspondence-aware and surface-only retrieval. For $D_0$–$D_2$, full-statement LLM selection recovers the relevant demonstration and reproduces oracle-level ICL performance, indicating that the paired ICL result is not an artifact of manual pairing in this few-shot setting and that oracle retrieval is attainable at near distances in this 64-seed pool. For $D_3$–$D_4$, the desired relation is no longer textual equivalence but algorithmic correspondence, which current retrieval strategies do not reliably identify; this defines the practical bottleneck for ICL beyond near transfer.

## G. Robustness and Ablations

### G.1. Algorithm and Regularization Variant Comparison

We compare SFT regularization variants to justify the supervised baseline used in the main experiments.

**SFT variants: Vanilla vs. GEM.** Table 11 compares vanilla SFT with SFT augmented by GEM (Gradient Equilibrium Method) regularization. While GEM is designed to improve generalization by balancing gradient contributions, we find that vanilla SFT achieves faster convergence on $D_0$ in our setting. Both methods show similar generalization profiles on $D_1$–$D_4$ when compared at matched memorization levels; we therefore report vanilla SFT in the main text for simplicity.

*Table 11.* SFT variant comparison on Qwen3-4B-Thinking. Pass@1 at step 500 (approximately matched $D_0$).

| Method | D0 | D1 | D2 | D4 |
|---|---|---|---|---|
| SFT (lr=2e-5) | 0.50 | 0.36 | 0.14 | 0.28 |
| SFT (lr=1e-5) | 0.37 | 0.32 | 0.13 | 0.30 |
| GEM (lr=2e-5) | 0.30 | 0.25 | 0.13 | 0.05 |
| GEM (lr=5e-6) | 0.23 | 0.18 | 0.08 | 0.05 |

## G.2. Extended Spectrum Profiles for All Variants

This appendix extends Table 3 with additional trainable methods under the same matched-memorization protocol. Rows are grouped by $D_0$ buckets and report pass@1 across $D_0$–$D_4$. The added methods are self-distillation (SDFT, §5.3), hint-assisted GRPO (Hint GRPO-Pseudo / Hint GRPO-KI, §5.4), and off-policy SFT with GPT-OSS-20B targets (§5.2). The self-generated-target SFT condition is reported as RFT and is therefore not duplicated. Each bucket lists the same trainable methods; in metric columns, "—" indicates that no checkpoint for that method reached the corresponding memorization bucket with a complete $D_0$–$D_4$ profile under the training budget.

*Table 12.* Extension of Table 3: pass@1 across the Generalization Spectrum for methods evaluated under the matched-memorization protocol.

| Method | Step | $D_0$ | $D_1$ | $D_2$ | $D_3$ | $D_4$ |
|---|---|---|---|---|---|---|
| Base model | — | 0.26 | 0.24 | 0.29 | 0.24 | 0.22 |
| ICL (paired, oracle) | — | 0.65 | 0.57 | 0.63 | 0.17 | 0.10 |
| ICL (random) | — | 0.21 | 0.17 | 0.21 | 0.06 | 0.10 |
| *Matched $D_0 \approx 0.84$* | | | | | | |
| SFT | — | — | — | — | — | — |
| RFT | 800 | 0.84 | 0.46 | 0.32 | 0.30 | 0.31 |
| DAPO | — | — | — | — | — | — |
| GRPO | — | — | — | — | — | — |
| SDFT | — | — | — | — | — | — |
| Hint GRPO-Pseudo | — | — | — | — | — | — |
| Hint GRPO-KI | — | — | — | — | — | — |
| Off-policy SFT (GPT-OSS) | — | — | — | — | — | — |
| *Matched $D_0 \approx 0.8$* | | | | | | |
| SFT | — | — | — | — | — | — |
| RFT | — | — | — | — | — | — |
| DAPO | 370 | 0.80 | 0.74 | 0.56 | 0.34 | 0.42 |
| GRPO | 290 | 0.80 | 0.72 | 0.49 | 0.34 | 0.37 |
| SDFT | — | — | — | — | — | — |
| Hint GRPO-Pseudo | — | — | — | — | — | — |
| Hint GRPO-KI | — | — | — | — | — | — |
| Off-policy SFT (GPT-OSS) | — | — | — | — | — | — |
| *Matched $D_0 \approx 0.7$* | | | | | | |
| SFT | 1900 | 0.69 | 0.44 | 0.29 | 0.18 | 0.20 |
| RFT | 500 | 0.71 | 0.40 | 0.31 | 0.31 | 0.33 |
| DAPO | 230 | 0.71 | 0.63 | 0.44 | 0.33 | 0.36 |
| GRPO | 220 | 0.71 | 0.63 | 0.43 | 0.32 | 0.34 |
| SDFT | — | — | — | — | — | — |
| Hint GRPO-Pseudo | — | — | — | — | — | — |
| Hint GRPO-KI | 110 | 0.70 | 0.63 | 0.42 | 0.21 | 0.30 |
| Off-policy SFT (GPT-OSS) | — | — | — | — | — | — |
| *Matched $D_0 \approx 0.6$* | | | | | | |
| SFT | 900 | 0.61 | 0.41 | 0.31 | 0.19 | 0.22 |
| RFT | 250 | 0.60 | 0.40 | 0.34 | 0.29 | 0.31 |
| DAPO | 180 | 0.60 | 0.57 | 0.42 | 0.29 | 0.35 |
| GRPO | 170 | 0.60 | 0.54 | 0.38 | 0.33 | 0.35 |
| SDFT | 60 | 0.61 | 0.51 | 0.40 | 0.13 | 0.16 |
| Hint GRPO-Pseudo | 70 | 0.60 | 0.57 | 0.38 | 0.23 | 0.27 |
| Hint GRPO-KI | 60 | 0.60 | 0.56 | 0.35 | 0.22 | 0.29 |
| Off-policy SFT (GPT-OSS) | 400 | 0.60 | 0.28 | 0.23 | 0.17 | 0.14 |

**Table 12** Extension of Table 3 (continued).

| Method | Step | $D_0$ | $D_1$ | $D_2$ | $D_3$ | $D_4$ |
|---|---|---|---|---|---|---|
| *Matched $D_0 \approx 0.5$* | | | | | | |
| SFT | 500 | 0.51 | 0.40 | 0.32 | 0.21 | 0.26 |
| RFT | 200 | 0.50 | 0.36 | 0.32 | 0.31 | 0.35 |
| DAPO | 120 | 0.49 | 0.47 | 0.38 | 0.29 | 0.33 |
| GRPO | 120 | 0.51 | 0.46 | 0.37 | 0.33 | 0.30 |
| SDFT | 40 | 0.51 | 0.47 | 0.37 | 0.14 | 0.16 |
| Hint GRPO-Pseudo | 40 | 0.54 | 0.51 | 0.36 | 0.25 | 0.28 |
| Hint GRPO-KI | 40 | 0.49 | 0.48 | 0.35 | 0.23 | 0.26 |
| Off-policy SFT (GPT-OSS) | 300 | 0.48 | 0.27 | 0.23 | 0.18 | 0.14 |
| *Matched $D_0 \approx 0.4$* | | | | | | |
| SFT | 100 | 0.42 | 0.41 | 0.29 | 0.19 | 0.19 |
| RFT | 100 | 0.38 | 0.34 | 0.31 | 0.30 | 0.35 |
| DAPO | 80 | 0.42 | 0.36 | 0.34 | 0.25 | 0.28 |
| GRPO | 60 | 0.41 | 0.34 | 0.32 | 0.24 | 0.29 |
| SDFT | 20 | 0.46 | 0.44 | 0.29 | 0.12 | 0.12 |
| Hint GRPO-Pseudo | 20 | 0.40 | 0.39 | 0.31 | 0.27 | 0.24 |
| Hint GRPO-KI | 30 | 0.44 | 0.45 | 0.32 | 0.24 | 0.27 |
| Off-policy SFT (GPT-OSS) | 100 | 0.40 | 0.25 | 0.23 | 0.18 | 0.15 |

For off-policy SFT, $D_2$ uses the `level2_v0` synthetic set used by the staged appendix training-run data. $D_3$ was rerun for the matched $D_0 \approx 0.4$, $0.5$, and $0.6$ checkpoints. Higher buckets remain "—" because the available run does not reach them with a complete $D_0$–$D_4$ profile.

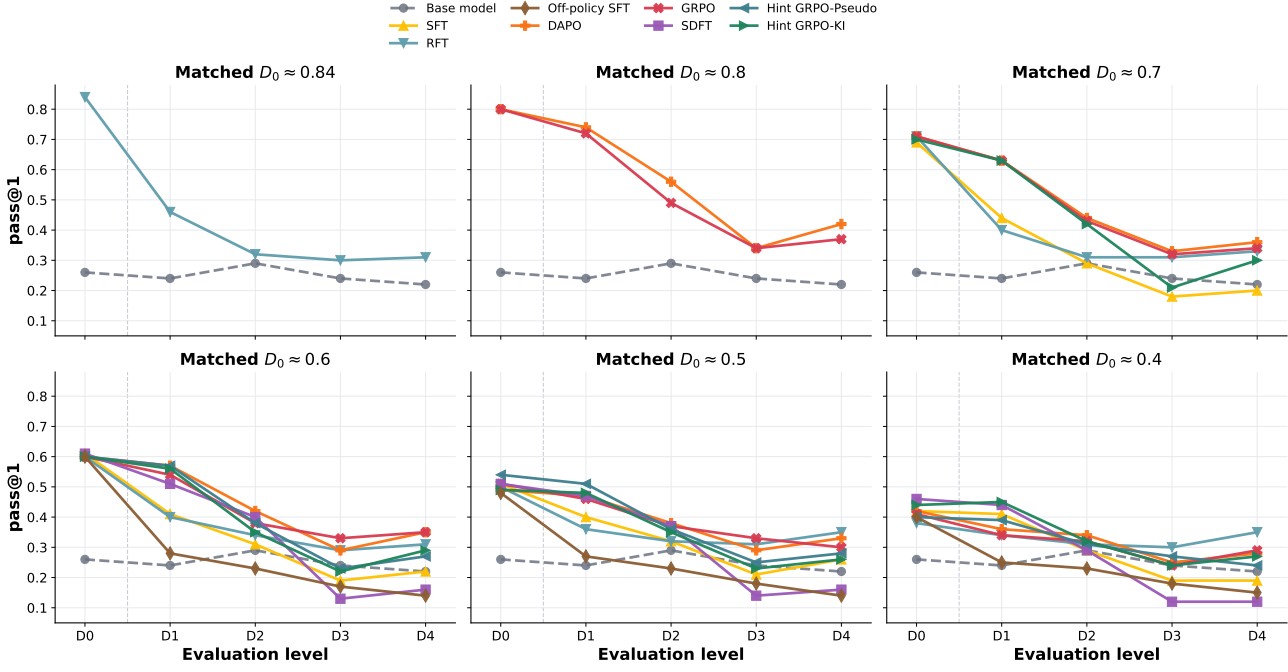

*Figure 10.* Generalization profiles at matched memorization levels. Each panel fixes a $D_0$ bucket and plots pass@1 across $D_0$–$D_4$ for methods with available entries in Table 12. The dashed gray line is the base model reference.

### G.3. Model Family Ablation

We replicate the full experimental protocol on DeepSeek-R1-Distill-Llama-8B, which differs from the main paper's Qwen3-4B-Thinking in architecture (Llama vs. Qwen), parameter count (8B vs. 4B), and training recipe (distillation from DeepSeek-R1 vs. native reasoning training). We apply the same seed selection, variant construction, and evaluation pipeline. Table 13 reports performance across the Generalization Spectrum.

*Table 13.* DeepSeek-R1-Distill-Llama-8B across the Generalization Spectrum. $\text{Gain}_n$ columns report normalized gains, AUS reports average raw transfer gain, and $\text{N-F}_n$ reports the normalized near-far gap. Format follows Table 3.

| Method | $D_0$ Result | $D_0$ $\text{Gain}_n$ | $D_1$ Result | $D_1$ $\text{Gain}_n$ | $D_2$ Result | $D_2$ $\text{Gain}_n$ | $D_3$ Result | $D_3$ $\text{Gain}_n$ | $D_4$ Result | $D_4$ $\text{Gain}_n$ | **AUS** | **$\text{N-F}_n$** |
|---|---|---|---|---|---|---|---|---|---|---|---|---|
| Base model | 0.18 | — | 0.02 | — | 0.16 | — | 0.12 | — | 0.03 | — | 0.00 | 0.0 |
| ICL (oracle) | 0.66 | +58.8 | 0.57 | +56.2 | 0.63 | +56.3 | 0.11 | $-1.3$ | 0.13 | +10.3 | +0.28 | +51.8 |
| SFT ($D_0 \approx 0.5$) | 0.53 | +43.1 | 0.06 | +4.5 | 0.20 | +4.4 | 0.11 | $-1.8$ | 0.07 | +3.9 | +0.03 | +3.4 |
| RL ($D_0 \approx 0.5$) | 0.50 | +39.7 | 0.13 | +10.5 | 0.25 | +10.7 | 0.18 | +7.0 | 0.09 | +5.8 | +0.08 | +4.2 |
| SFT ($D_0 \approx 0.37$) | 0.37 | +23.9 | 0.05 | +3.2 | 0.17 | +0.2 | 0.11 | $-1.8$ | 0.04 | +1.2 | +0.01 | +2.0 |
| RL ($D_0 \approx 0.37$) | 0.37 | +23.7 | 0.09 | +6.6 | 0.23 | +8.2 | 0.15 | +2.9 | 0.05 | +2.2 | +0.05 | +4.9 |

The overall decay pattern replicates: (1) transfer gains attenuate from near to far levels for gradient-based methods; (2) RL's near-transfer advantage persists at matched $D_0$; (3) far-transfer results converge at $D_3$–$D_4$.

## G.4. Seed Set Scale Ablation

We expand the seed set from 64 to 256 by sampling an additional 192 problems from the same source (recent Codeforces problems), using the same filtering criteria as in the main study: the base model's solve rate is below 0.6 and pass@8 $> 0$. For each new seed, we construct the full $D_0$–$D_4$ variants following the main protocol, yielding 1,280 evaluation instances in total (256 seeds $\times$ 5 levels). We retrain both SFT and RL (GRPO) models on the expanded dataset with identical hyperparameters to the main experiments.

Figure 5 reports the pass@1 profile for this ablation. This appendix records the construction and training scope for that result.

