# OpenReview forum: "The Generalization Spectrum: A Chromatographic Approach to Evaluating Learning Algorithms"
_ICML.cc/2026/Conference — ICML 2026 regular_

### Official Review · Reviewer_EgVT · 2026-03-11

**Soundness:** 2
**Presentation:** 3
**Significance:** 3
**Originality:** 3
**Overall Recommendation:** 4
**Confidence:** 4

**Summary:**

The authors propose the Generalization Spectrum, a framework for evaluating generalization as a function of transfer distance rather than a binary in/out split. Essentially, paired test variants are constructed by deriving samples from the same seed instance, ordered by increasing distance/difficulty from the original. Experiments are designed to reveal the effect of transfer distance on generalization depending on one out of four common training scenarios: RL converts memorization into near-transfer more effectively than SFT and ICL, while on-policy solutions preserve transfer whereas off-policy supervision collapses performance toward memorization.

**Compliance With Llm Reviewing Policy:**

Affirmed.

**Key Questions For Authors:**

- How were the interpretations of failure modes and differences between models analyzed (Section 4.3, Section 5)? (see also Weaknesses)
- Do you have any insight on how different learning algorithms lead to different (internal) representations and/or algorithmic and compositional capabilities? See, e.g., [Sha25, Ebe25].

**Related Works**
- [Sha25] Sharkey, L, et al. Open Problems in Mechanistic Interpretability. TMLR 2025.
- [Ebe25]  Eberle, O., et al. Position: We Need An Algorithmic Understanding of Generative AI. ICML 2025.

**Limitations:**

Limitations are discussed in Section 7.

**Strengths And Weaknesses:**

**Strengths**
- The paper focuses on a relevant challenge of comparing learning paradigms in machine learning, essential for transfer learning and generalization.
- Well thought-out and sound experimental setup, following clearly defined desiderata and introducing plausible “Spectrum Levels.”
- Proposal of a Generalization Spectrum Benchmark, useful for further progress in this direction.
- Interesting discussion and analysis of amplified transfer rather than general
capability in ICL settings (Section 5.1)

**Weaknesses**
- Empirical focus on one task, i.e. competitive programming, which presents a limited use case.
- Failure modes and distinct failure clusters discussed in Section 4.3 and Problem 1860; Appendix B are not shown, limiting the depth and scientific rigor of the analysis. The authors also do not clearly state how error modes were analyzed, e.g., human inspection, LLM-based analysis, combined heuristics, etc. As a result, findings in Section 4.3 are not fully supported and not reproducible at this stage.
- Lack of theoretical analysis or discussion that could support some of the identified differences.
- All experiments are conducted using a single model (Qwen3-4B-Thinking), which limits the generalizability of the findings to other models.

---

> ### Author Rebuttal · Authors · 2026-03-31
>
> We thank the reviewer for the thorough evaluation and constructive suggestions.
>
> **W1: Single task.**
>
> We address domain scope in our response to Reviewer Uini (W1), where we discuss how the framework can extend and provide a concrete instantiation for mathematical reasoning.
> We chose competitive programming for its clean verification setup (§A.1), not as the only valid domain.
>
> **Q1 & W2: Failure modes methodology unclear.**
>
> Full details for each analysis:
>
> **D1 error classification (Figure 2, §4.3).** D1 failures were labeled automatically from compiler/runtime output: compile errors are flagged by the compiler, runtime errors by non-zero exit codes, and wrong answers by test-case mismatches. No manual judgment is used.
>
> **D2 error taxonomy (§4.3).** Our analysis followed a two-stage process:
>
> *Stage 1: Taxonomy construction.* Informed by prior error taxonomies for LLM-generated code [1,2], we manually examined ~50 wrong D2 solutions from both SFT and RL. Since prior categories did not fully fit context-transfer failures, we refined the taxonomy into four types: Algorithm mismatch (38%), Constraint omission (31%), Structure confusion (24%), and Other error (7%).
>
> *Stage 2: LLM-aided labeling.* We sampled 32 D2 problems, each with 8 attempts (512 solutions in total). GPT-5 labeled each wrong solution using a structured prompt with the original problem, gold solution, model code, and test pass rate. It returned JSON labels plus a short reason (temp=0.3). We spot-checked ~50 cases and found 86% agreement; disagreements were mostly between Constraint omission and Structure confusion, which naturally overlap.
>
> **Case studies (Problems 1860, 14816).** Beyond the systematic classification, we manually compared D0 and D2 solution pairs to build qualitative understanding. These examples were selected because they cleanly illustrated patterns that recurred across problems, e.g., the technique abandonment shown in Problem 14816 appeared in 12/18 (67%) of problems where SFT succeeded on D0 but failed on D2.
>
> We will include actual model outputs and additional examples, and release the full classification script, prompts, and representative labeling outputs with the revision.
>
> **W3: Lack of theoretical analysis.**
>
> Thank you for this thoughtful point. We'd like to note that our primary contribution is the evaluation framework and benchmark, and its design is grounded in principled methodology: the spectrum levels (Tab 1) systematically remove shared structure, and matched-memorization comparison functions analogously to controlling for confounds in causal inference. We think this gives strong empirical support.
>
> That said, our related work (§6) already connects to several lines of theoretical and empirical analysis that support our findings—for example, [3] provide evidence that SFT memorizes while RL generalizes, [4] show improved robustness for RL under distribution shift, and [5] analyze the bounded OOD generalization of ICL. Our framework provides finer-grained behavioral evidence for these claims by decomposing "generalization" into graded transfer distances rather than binary splits. In the revision, we will expand this discussion to more explicitly connect our empirical observations (e.g., SFT's surface binding in Tab 4, ICL's divergent transfer mechanism at D2) to the theoretical explanations offered by these works, making the conceptual grounding more prominent.
>
> **W4: Single model.**
>
> We now address this with new experiments on DeepSeek-R1-Distill-Llama-8B. Full results and analysis appear in our reply to Reviewer EPJc (W1) due to space limits. The main patterns hold across a different architecture, scale, and training recipe, which suggests they reflect broader algorithmic effects rather than model-specific artifacts.
>
> **Q2: Insight on internal representations? [Sha25, Ebe25]**
>
> Thank you for these references. We see them as highly complementary.
>
> Our finding that "SFT learns techniques while RL learns properties" (Tab 4) provides behavioral evidence for the algorithmic divergence that Eberle et al. [Ebe25] argue should be studied systematically. Our D2 level acts as a natural test of surface-bound vs structure-bound representations, paralleling AlgEval's goal. Meanwhile, Sharkey et al. [Sha25] identify "predicting capabilities that arise during finetuning" as a key open problem; Our matched-memorization protocol and graded spectrum levels provide the kind of behavioral measurement setup needed to test such mechanistic ideas. We will add discussion in the revision.
>
> [1] What's Wrong with Your Code Generated by Large Language Models?
> [2] Evaluating and Improving LLM-based Competitive Program Generation.
> [3] Sft memorizes, rl generalizes: A comparative study of foundation model post-training.
> [4] Understanding the effects of RLHF on LLM generalisation and diversity.
> [5] Can in-context learning really generalize to out-of-distribution tasks?

---

> > ### Author Rebuttal · Reviewer_EgVT · 2026-04-01
> >
> > Thank you for the detailed response. My points and suggestions are adequately addressed; including the provided details, changes, complementary related work and additional results will strengthen the paper. I will maintain my positive assessment.

---

> > > ### Author Response · Authors · 2026-04-03
> > >
> > > Thank you for your positive reassessment and for the pointers to [Sha25] and [Ebe25], which genuinely enriched our discussion. We will incorporate the additional details on failure-mode methodology, model outputs, and the expanded theoretical connections in the revision. We're grateful for your thorough and constructive review!

---

### Official Review · Reviewer_EPJc · 2026-03-11

**Soundness:** 3
**Presentation:** 2
**Significance:** 2
**Originality:** 3
**Overall Recommendation:** 4
**Confidence:** 3

**Summary:**

A simple metric, such as average accuracy on a dataset, misses a lot of interesting substructure in the performance of LLMs. This paper breaks down tasks an LLM can perform into 5 different categories, which have intuitively different requirements for generalization for the model to perform. This begins with simple memorization, all the way to performing in-distribution tasks that are not simply recontextualizing something the model already knew. This hierarchy is empirically defended by seeing models performing low categories on this ladder (D_0) with much more ease than those further on the ladder (D_3). This work uses this to show that RL shows better generalization on harder problems for the same performance on easier-to-generalize problems as SFT.

**Compliance With Llm Reviewing Policy:**

Affirmed.

**Final Justification:**

The revisions described in the rebuttal introduce two important additions to this work: the extension of the spectrum beyond solely competitive programming and the incorporation of a second model to evaluate results across multiple architectures. These additions strengthen the significance of the work, as the findings are no longer contingent on a narrow range of tests.

Although the clarity of the submitted draft is somewhat difficult to follow, the authors’ responses to my review and those of other reviewers indicate that revisions have been made to improve clarity. The experiments and results are both broad and deep, showing that the observed behavior is not a result of surface-level differences such as length, as noted by reviewer Uini.

This work does, however, only test the one setting of competitive programming and not other settings, such as mathematical reasoning. Competitive programming provides a strong testbed for the spectrum, but it may only be well-suited for competitive programming empirically, without additional results.

Overall, the proposed spectrum provides a clear and well-structured progression for measuring the performance of different models within the context of structured problems such as solving programming challenges. Others likely can adopt this framework to other task settings, broadening the applicability.

**Key Questions For Authors:**

1. Can these results be seen in a wider array of models? For example, similar differences between the performances of the different categories, as seen with Qwen, for the other models here, such as GPT-OSS and QWQ, would strengthen the generality of the claims made. The failure analysis for D1 and D2 for SFT and RL is quite interesting, but it's limited in its scope as is.
2. Is there a difference between GG and MG other than that $i=0$ for MG and $i \not = 0$ for GG? The difference in name makes reading tables with different metrics confusing, even though they are identical in form.
3. Section 2.3 also mentions this is source agnostic, not just model agnostic. Was this framework tested on other sources? Can "sources" include problems that are not competitive programming?

**Limitations:**

yes

**Strengths And Weaknesses:**

Strengths
- Creating datasets with a progressively more complex collection of tasks to test general LLMs on provides a much finer-grained look into the differences between highly-performing LLMs.
- These categories gave much more detailed separations when looking at differences between learning algorithms, from implicit ones such as ICL to RL and SFT
- This categorization also quantifies how models with stronger abilities to memorize tend to perform better with RL. This and other properties are nicely exposed by this characterization.

Weaknesses
- For a paper centering on empirical results for a gathered collection of tasks, using mainly Qwen3 for the results leaves the possibility of these effects mainly being a result of Qwen3. Two models, GPT-OSS and QWQ, are used, but only in a pair of experiments.
- The category definitions are not written out very clearly. The single sentences in section 2.1 are somewhat vague, and there are no examples of each category until the appendix (if there is a reference to the appendix to see examples, it was missed)
- The different categories are hard to follow and appear fairly orthogonal. For example, D1 is interested in differences in implementation language, while D2 is interested in changing the problem setup to be equivalent but written differently. There wasn't a clear reason as to why there would be an increase in difficulty in the tasks, other than what seems to empirically fit.

---

> ### Author Rebuttal · Authors · 2026-03-31
>
> We thank the reviewer for the detailed feedback. We address each concern below.
>
> **W1 & Q1: Limited model families.**
>
> We agree this is important. We have conducted new experiments on DeepSeek-R1-Distill-Llama-8B (Llama), which differs from Qwen3-4B-Thinking in architecture (Llama vs. Qwen), parameter count (8B vs. 4B), and training recipe (distillation from DeepSeek-R1 vs. native reasoning training). We apply the identical experimental protocol.
>
> **Tab R1.** Llama across the Generalization Spectrum (format follows Tab 3). Gain(i) = r_i(M_S) − r_i(M), normalized by remaining headroom.
>
> | Method | D0 (Exact Recall) | | D1 (Impl. Transfer) | | D2 (Context Trans.) | | D3 (Category Match) | | D4 (Unpaired) | | AUS | N-F |
> |-|-|-|-|-|-|-|-|-|-|-|-|-|
> | | Result | Gain(0) | Result | Gain(1) | Result | Gain(2) | Result | Gain(3) | Result | Gain(4) | | |
> | Base model | 0.18 | — | 0.02 | — | 0.16 | — | 0.12 | — | 0.03 | — | 0.00 | 0.00 |
> | ICL (oracle) | 0.66↑+.48 | +58.8% | 0.57↑+.55 | +56.2% | 0.63↑+.47 | +56.3% | 0.11↓-.01 | -1.3% | 0.13↑+.10 | +10.3% | +0.28 | +0.47 |
> | SFT (D0≈0.5) | 0.53↑+.35 | +43.1% | 0.06↑+.04 | +4.5% | 0.20↑+.04 | +4.4% | 0.11↓-.02 | -1.8% | 0.07↑+.04 | +3.9% | +0.03 | +0.03 |
> | RL (D0≈0.5) | 0.50↑+.33 | +39.7% | 0.13↑+.10 | +10.5% | 0.25↑+.09 | +10.7% | 0.18↑+.06 | +7.0% | 0.09↑+.06 | +5.8% | +0.08 | +0.04 |
> | SFT (D0≈0.37) | 0.37↑+.20 | +23.9% | 0.05↑+.03 | +3.2% | 0.17↑+.00 | +0.2% | 0.11↓-.02 | -1.8% | 0.04↑+.01 | +1.2% | +0.01 | +0.02 |
> | RL (D0≈0.37) | 0.37↑+.20 | +23.7% | 0.09↑+.06 | +6.6% | 0.23↑+.07 | +8.2% | 0.15↑+.03 | +2.9% | 0.05↑+.02 | +2.2% | +0.05 | +0.04 |
>
> Key findings replicate: (1) monotonic Gain(i) decay from D1→D4 for all gradient-based methods; (2) RL's near-transfer advantage persists at matched D0; (3) far-transfer convergence at D3–D4; (4) SFT surface-binding replicates (see Tab R2).
>
> We extend the failure analysis to Llama. Note that this model has weak baseline C++ competence (52.9% compile error rate even before training), making the absolute rates higher than Qwen3; the key diagnostic is the *relative* divergence between SFT and RL.
>
> **Tab R2.** Compile error rates on D1 for Llama.
> | Method | Compile Error Rate (D1) |
> |-|-|
> | Base model | 52.9% |
> | SFT (D0≈0.37) | 68.6% |
> | SFT (D0≈0.5) | 77.7% |
> | RL (D0≈0.37) | 59.4% |
> | RL (D0≈0.5) | 66.4% |
>
> SFT sharply *increases* compile errors over the base model (+15.7pp at D0≈0.37, +24.8pp at D0≈0.5), while RL's increase is substantially smaller (+6.5pp, +13.5pp). This replicates the main paper's finding: SFT binds to Python surface patterns and corrupts C++ generation, whereas RL better preserves cross-lingual competence. The pattern holds on this architecturally distinct model, confirming surface-binding is a property of the learning algorithm. We will provide the corresponding D2 failure analysis for Llama in the next revision round. Together with Qwen3, our framework is validated across two model families spanning 4B–8B parameters.
>
> **Q2: Difference between GG and MG.**
>
> MG and GG share the same form (r_i(M_S)−r_i(M)), differing only in which level they apply to (i=0 for MG, i≥1 for GG). We agree two names adds cognitive load. In the revision, we will unify notation using Gain(i) and use "memorization gain" and "generalization gain" only as verbal shorthands when the distinction matters.
>
> **Q3: Source-agnostic claim; applicability beyond competitive programming.**
>
> "Source-agnostic" refers to data collection sources: our seeds aggregate from RStar-Coder, CodeContestPlus, and LiveCodeBench-Pro, each independent.
>
> The core principle—constructing paired variants by progressively stripping shared information—is domain-agnostic. For math: D0=original; D1=changed variables/values; D2=different scenario preserving structure; D3=same technique family; D4=unrelated in-domain. Recent single-level math perturbations [1,2] suggest building blocks exist. We chose competitive programming for clean verification (§A.1) and leave multi-domain extension to future work.
>
> **W2: Category definitions unclear.**
>
> We will expand each level's description in §2.1 with a concrete example and add the missing cross-reference to Appendix B.
>
> **W3: Categories appear orthogonal; difficulty ordering unclear.**
>
> The spectrum measures distance from D0 in shared information, not task difficulty. Each level is defined by what it shares with the seed (Tab 1): D1 shares statement+solution+family; D2 shares solution+family; D3 shares only family; D4 shares nothing. We acknowledge D1 and D2 are conceptually distinct perturbation types; we project them onto a single axis to measure cumulative information loss. Tab 3 shows this ordering surfaces meaningful differences (e.g., ICL vs. gradient-based divergence at D1 vs. D2). Quantitative validation (cosine similarity and tag overlap) is in our response to Reviewer 3RwU (Q3); we will revise §2.1 accordingly.
>
> [1] MATH-Perturb.
> [2] VeRA.

---

> > ### Author Rebuttal · Reviewer_EPJc · 2026-04-02
> >
> > I thank the authors for their detailed responses. The additional experiment involving Llama improves the breadth of the work.
> >
> > Most importantly, my understanding was significantly improved by the clarification that the proposed framework is not critically tied to competitive programming. Section 2.1 previously indirectly suggested that the framework was specific to this setting, even though Table 1 reflected the underlying nesting structure. This created the impression that the categories were orthogonal, when in fact they are related in an important and meaningful way.
> >
> > With the planned revisions to Section 2.1, the work demonstrates novelty in establishing a highly controlled setting for measuring generalization, including applications to other tasks of interest to the broader research community. In light of these clarifications and improvements, I have raised my score accordingly.

---

> > > ### Author Response · Authors · 2026-04-03
> > >
> > > Thank you for re-evaluating our work and for raising your score! Your valuable feedback on expanding model coverage and clarifying Section 2.1 directly improved the paper. The Llama experiments and the revised category definitions will both appear in the updated manuscript. We appreciate the constructive dialogue so much!

---

### Official Review · Reviewer_Uini · 2026-03-12

**Soundness:** 4
**Presentation:** 3
**Significance:** 3
**Originality:** 3
**Overall Recommendation:** 5
**Confidence:** 5

**Summary:**

This paper examines how different learning algorithms generalise. To do so, they define a spectrum of levels where the problems increasingly deviate from the training ones. They define metrics to measure memorisation and generalisation, and evaluate the performance of ICL, SFT, RFT, and RL using a competitive programming benchmark. Results show that generalisation varies across algorithms and transfer levels. They also show the problems methods face across different spectrum levels. Finally, they investigate how variations in the content design affect performance.

**Compliance With Llm Reviewing Policy:**

Affirmed.

**Final Justification:**

The authors have clarified several points and committed to improving parts of the paper that required further explanation. One of the main weaknesses regarding model diversity has now been addressed, with results provided for an additional model with a different architecture, which strengthens the empirical evaluation.

I would still be interested in seeing the method applied to another domain. However, I understand that this is likely out of scope for the current work, and the authors have reasonably argued that the approach could be extended straightforwardly.

**Key Questions For Authors:**

1. How would the framework apply to other domains? How would the spectrum levels be defined? Could the authors provide an example?
2. In Table 2, what does Range stand for? Also, why are there fewer algorithm categories in D4?
3. How is the model error clustering done on D2?
4. Table 4 is not referenced. Could the authors clarify the conclusions of this table and what it adds to the results?

**Limitations:**

yes

**Strengths And Weaknesses:**

Strengths
- The evaluation framework the authors propose is novel and interesting. It provides a way to actually measure generalisation across different levels, rather than comparing methods using an aggregate metric. This ensures a more comprehensive comparison.
- The experimental design is sound; they not only show surface-level results, but they also inspect problem size, investigate why the methods might not work, and change content design to explore possible improvements.
- The presentation is good.

Weaknesses
- The work has its main limitation in that, even though it is extensive, it is applied only to one domain, leaving open the question of whether it can be easily applied to others.
- Even if they use different learning algorithms, the base model is the same. I have concerns if the results could change with a different model.
- The paper would benefit from a more detailed method explanation before the results.

---

> ### Author Rebuttal · Authors · 2026-03-31
>
> We thank the reviewer for the positive assessment and thoughtful questions.
>
> **W1: More domains.**
>
> Thank you for this helpful suggestion. While we instantiate the framework on competitive programming in this paper, the core design principle is domain-agnostic: each level is defined by progressively removing information shared between the seed and test problem. This same logic can be applied beyond code tasks.
>
> For mathematical reasoning, for example, one can define: D0 as the original problem; D1 as variants with changed variable names or numerical values; D2 as a different application scenario that preserves the same underlying mathematical structure; D3 as a different problem requiring the same technique family; and D4 as unrelated in-domain problems. Concretely, a problem about counting integer solutions to x²+y²≤n could be reframed at D2 as counting lattice points inside a circle, while preserving the same core solution structure.
>
> We chose competitive programming as our first testbed because it provides clean and scalable verification through executable test cases (§A.1), which makes matched-memorization comparison especially reliable. We agree that extending the spectrum to other reasoning domains is an important next step, and we will clarify this broader applicability in the revision.
>
> **W2: More models.**
>
> We have conducted new experiments on DeepSeek-R1-Distill-Llama-8B (Llama-3.1-8B-Base distilled from DeepSeek-R1)--a different architecture, scale, and training recipe from Qwen3-4B-Thinking. Full results are in our response to Reviewer EPJc (W1) due to the space limits. We observe consistent patterns: monotonic Gain(i)/GGₙ decay, RL's near-transfer advantage at matched D0, and convergence in far transfer.
>
> **W3: More detailed method explanation before results.**
>
> We appreciate this suggestion. In the revised manuscript, we will add an "Evaluation Pipeline" summary at the end of Section 3.3: (1) apply each method (ICL/SFT/RFT/RL) on the same 64 D0 seeds; (2) evaluate D0 checkpoints to identify matched-memorization points (e.g., both SFT and RL at pass@1≈0.8); (3) freeze these checkpoints and evaluate on D1–D4; (4) compute GGₙ at each level to quantify transfer extent. This clarifies that all comparisons are conditioned on equivalent memorization.
>
> **Q2a: What does "Range" stand for in Table 2?**
>
> "Range" refers to the min–max span of Codeforces difficulty ratings across problems at each level (e.g., 900–2600 for D0/D1). We will revise the caption to make it clear.
>
>
> **Q2b: Why are there fewer algorithm categories in D4?**
>
> D4 problems are randomly sampled with no category-matching constraint (unlike D3, which enforces shared tags via bipartite matching). The difference (23 vs 27 categories) is a natural consequence of drawing from a finite pool. We will add: "D4 covers 23 of 27 categories due to unconstrained random sampling; the slightly reduced coverage does not affect its role as an unpaired baseline."
>
> **Q3: How is the model error clustering done on D2?**
>
> Thank you for this question. We used a two-stage procedure to cluster D2 errors.
>
> First, we manually examined approximately ~50 incorrect D2 solutions from both SFT and RL checkpoints to construct a taxonomy tailored to context-transfer failures. Guided by prior work on LLM code errors [1,2], we identified four recurring categories: Algorithm mismatch (38%), Constraint omission (31%), Structure confusion (24%), and Other error (7%).
>
> Second, we scaled this analysis using GPT-5 with a structured classification prompt. We sampled 32 D2 problems per method, with 8 attempts per problem (approximately 512 solutions per method). For each incorrect solution, the classifier was given the original problem, ground-truth solution, the model's code, and its test pass outcome, and returned a JSON-formatted label plus justification.
>
> To verify reliability, we manually spot-checked ~50 model-labeled cases and observed 86% agreement; most disagreements were between *Constraint omission* and *Structure confusion*, which are naturally adjacent categories. In the revision, we will add the full classification protocol, aggregate statistics, and representative examples to Appendix B.
>
> **Q4: Table 4 is not referenced.**
>
> Thank you for catching this. Table 4 summarizes the key finding from Section 4.3: SFT learns surface-bound techniques (e.g., 2D prefix sums) that fail to transfer under narrative change, while RL learns structure-bound properties (e.g., geometric bounding box) that transfer intact. We will add an explicit reference in the text.
>
> [1] MATH-Perturb: Benchmarking LLMs' Math Reasoning Abilities against Hard Perturbations.
> [2] VeRA: Verified Reasoning Data Augmentation at Scale Human-Free Verification for Boundary-Aware Evaluation of Frontier Reasoning Models.
> [3] What's Wrong with Your Code Generated by Large Language Models?
> [4] Evaluating and Improving LLM-based Competitive Program Generation.

---

> > ### Author Rebuttal · Reviewer_Uini · 2026-04-03
> >
> > I thank the authors for the comprehensive rebuttal. All my concerns have been addressed. Although I would like to see how the method works with other domains, I understand that it is extensive work, and I appreciate the example. I also really value the new experiments with the other model, and I have updated my score accordingly.

---

> > > ### Author Response · Authors · 2026-04-04
> > >
> > > Thank you so much for raising your score and for the thoughtful engagement throughout the review process! Your suggestions on extending to other domains, adding experiments with additional models and other clarifications were instrumental in strengthening the paper. We will include them all in the revision. We truly appreciate the constructive and encouraging dialogue!

---

### Official Review · Reviewer_3RwU · 2026-03-12

**Soundness:** 3
**Presentation:** 3
**Significance:** 3
**Originality:** 3
**Overall Recommendation:** 4
**Confidence:** 3

**Summary:**

This paper introduces the Generalization Spectrum, a framework for evaluating model generalization by measuring performance across increasing transfer distances between training and test examples. Using a competitive programming benchmark with paired problem variants, the authors compare several learning paradigms under a matched memorization protocol to isolate transfer efficiency. Results show that generalization drops as transfer distance increases. RL outperforms SFT/RFT on near transfer (D1–D2), while ICL works best for context transfer when demonstrations are well matched. The paper also finds that abstract demonstrations improve ICL and on-policy supervision preserves transfer better than off-policy SFT, showing that both the training method and supervision content influence generalization.

**Compliance With Llm Reviewing Policy:**

Affirmed.

**Final Justification:**

Thank you to the authors for their rebuttal. I believe it adequately addresses my concerns, and I am therefore happy to raise my rating to weak accept (4).

**Key Questions For Authors:**

Questions:
1- Do the authors expect the proposed Generalization Spectrum and the main empirical findings to transfer beyond competitive programming, for example to mathematical reasoning or natural language tasks?
2- How stable are the reported conclusions across different model families and parameter scales? In particular, do the advantages of RL on near-transfer and ICL on context transfer hold more broadly?
3- Can the authors provide additional validation that the paired problem variants at different spectrum levels consistently reflect the intended increase in transfer distance?
4- Since ICL performance appears to depend heavily on well-matched demonstrations, how would the results change under more realistic retrieval settings without oracle access to paired examples?

**Limitations:**

yes

**Strengths And Weaknesses:**

Strengths:
1- evaluation framework: The Generalization Spectrum offers a clearer way to separate memorization from transfer across controlled distances.
2- experimental methodology: The matched-memorization protocol makes comparisons between learning paradigms fairer and more convincing.
3- Useful empirical insights: The paper shows that training content matters, with abstract ICL demonstrations and on-policy supervision improving transfer.


Weaknesses:
1- All experiments are conducted on competitive programming, so it is unclear how well the findings generalize to other reasoning or language domains.
2- The main conclusions are drawn from experiments on a single base model, which makes it hard to know whether the observed patterns are specific to that model or reflect a broader phenomenon. Evaluating at least one additional model family would make the claims more convincing.
3- Some conclusions, especially for ICL, rely on paired/oracle-style retrieval and constructed problem variants, which may limit practical realism.
4- Appendix B, line 774 contains an unresolved reference (??), which should be fixed.

---

> ### Author Rebuttal · Authors · 2026-03-31
>
> We thank the reviewer for the careful reading and thoughtful questions.
>
> **W1 & Q1: Domain transferability.**
>
> We address this in our response to Reviewer Uini (W1) due to space limits, including a concrete D0-D4 setup for mathematical reasoning. We expect the key finding, RL turns memorization into near-transfer more efficiently than SFT, to hold wherever outcome-based reward is available, since the mechanism (§4.3, Table 4) is not programming-specific: SFT binds knowledge to surface features while RL learns structural properties that survive recontextualization. Competitive programming serves as a controlled testbed; confirming these findings across domains is a priority for future work.
>
> **W2 & Q2: Stability across model families and scales.**
>
> We agree and ran new experiments on DeepSeek-R1-Distill-Llama-8B, which differs from Qwen3-4B-Thinking in architecture (Llama vs. Qwen), scale (8B vs. 4B), and training recipe (Llama-3.1-8B-Instruct further distilled with DeepSeek-R1 reasoning data). Full results are in our response to Reviewer EPJc (W1) due to space limits. All core findings replicate: monotonic GGn decay, RL's near-transfer advantage over SFT/RFT at matched memorization, and convergence at far-transfer (D3-D4). We will add these results to the revision.
>
> **Q3: Validation that spectrum levels reflect intended transfer distance.**
>
> We validate this ordering through two complementary lenses:
>
> *1. Structural (by design).* Each level progressively removes shared information with D0 (Tab 1): D1 shares statement+solution+tests+family; D2 shares solution+tests+family; D3 shares family only; D4 shares nothing.
>
> *2. Quantitative similarity.* We computed cosine similarity (all-mpnet-base-v2 embeddings) between D0 seeds and paired variants, averaged over 64 pairs:
>
> | Level | Avg. Cosine Sim. | Std. Dev. |
> |-|-|-|
> | D1 | 1.000 | 0.000 |
> | D2 | 0.513 | ±0.086 |
> | D3 | 0.373 | ±0.095 |
> | D4 | 0.346 | ±0.096 |
>
> The D1→D2 drop (1.000→0.513) validates our recontextualization pipeline. D3 and D4 have similar text similarity because both use different statements, but they differ structurally: algorithmic tag overlap is 100% for D3 (64/64 share ≥1 tag with D0) and 17.2% for D4 (11/64, incidental).
>
> The convergence of both lenses provides strong evidence that spectrum levels reflect the intended distance ordering. We will add this analysis to revised Appendix A.
>
> **W3 & Q4: ICL under realistic (non-oracle) retrieval.**
>
> We agree that oracle retrieval is an upper bound and ran additional experiments using the same 64-seed training pool:
>
> - **Text Embedding**: E5-Mistral-7B-Instruct [1] embeds problem statements and retrieves the nearest neighbor by cosine similarity.
> - **Embed + LLM Rerank**: Top-20 by embedding, then reranked by GPT-5.4-thinking.
> - **LLM Selector**: All 64 seed statements (~28k tokens) in one context window, and GPT-5.4-thinking selects the most algorithmically relevant demonstration.
> - **Oracle** and **Random** as upper and lower controls.
>
> Tab 1. Retrieval Accuracy (Recall@1 for finding the true paired seed):
> | Retrieval Method | D0/D1 | D2 | D3 |
> |-|-|-|-|
> | Oracle | 100% | 100% | 100% |
> | LLM Selector | 100% | 100% | 0% |
> | Embed + LLM Rerank | 100% | 78.0% | 0% |
> | Text Embedding | 100% | 47.5% | 0% |
> | Random | 1.6% | 1.6% | 1.6% |
>
>
> Tab 2. ICL Performance under Realistic Retrieval:
> | Retrieval Method | D0 | D1 | D2 | D3 | D4 |
> |-|-|-|-|-|-|
> | Baseline | 0.26 | 0.24 | 0.29 | 0.24 | 0.22 |
> | Oracle | 0.65 | 0.57 | 0.63 | 0.17 | 0.10 |
> | LLM Selector | 0.65 | 0.57 | 0.64 | 0.12 | 0.16 |
> | Embed + LLM Rerank | 0.64 | 0.57 | 0.55 | 0.13 | 0.17 |
> | Text Embedding | 0.65 | 0.58 | 0.44 | 0.15 | 0.20 |
> | Random | 0.21 | 0.17 | 0.21 | 0.06 | 0.10 |
>
> As Tab. 1 shows, the LLM Selector already achieves 100% recall on D0–D2 when given full candidate statements (for reference, truncating to 240 characters drops D0 recall to 7.8%, showing the value of full context). Accordingly, Tab. 2 shows that the LLM Selector nearly matches oracle performance on D0–D2 (e.g., 0.64 vs. 0.63 on D2). More broadly, ICL performance tracks retrieval accuracy directly: on D2, pass@1 drops from 0.64 to 0.55 to 0.44 as recall drops from 100% to 78% to 47.5%. At D3-D4, since no retrieval method can identify structurally paired seeds (0% recall), no method produces meaningful ICL gains–consistent with our paper's conclusions. Together, these results confirm that oracle retrieval is a realistic assumption for D0-D2, and that the practical bottleneck for ICL lies in identifying algorithmic correspondence rather than surface similarity. We will add the full results in the revision.
>
> **W4: Unresolved reference.**
>
> Thank you for catching this. The reference in Appendix B (Line 774) should point to Section 4.3. We have corrected this.
>
> [1] Improving Text Embeddings with Large Language Models. ACL 2024.

---

> > ### Author Rebuttal · Reviewer_3RwU · 2026-04-04
> >
> > Thank the authors for their responses. My concerns have been addressed.

---

> > > ### Author Response · Authors · 2026-04-04
> > >
> > > Thank you for confirming that your concerns have been addressed! Would you be open to considering updating your score to reflect this? We really appreciate your constructive feedback!

---

### Decision · Program_Chairs · 2026-04-30

**Decision:**

Accept (regular)

**Comment:**

The submission proposes a so-called "chromatographic spectrum" for evaluating LLMs. This approach involves creating additional test data through carefully crafted augmentations that share varying levels of information with the original test instances. Rather than reporting performance on only the original test data, it is recommended to report multidimensional performance measured across these different types of augmentations. Experiments are conducted using an LLM on a dataset about competitive programming, where the performance of different learning paradigms (SFT, RL, etc) are compared using this framework.

The initial reviews found that the limitation of the experimental setup to a single dataset and language model were a cause for concern. I agree that this is a significant shortcoming, so the addition of the Llama experiments during the rebuttal period are welcome. Other concerns related to the presentation and justification for the relative difficulty of each test set variant were also noted as being resolved by the reviewers. Due to the outcome of the discussion between the authors and reviewers during the rebuttal phase, I tentatively recommend acceptance of the submission.